# Laboratory, field, mast-borne and airborne spectral reflectance measurements of boreal landscape during spring

Henna-Reetta Hannula[1], Kirsikka Heinilä[2], Kristin Böttcher[2], Olli-Pekka Mattila[3], Miia Salminen[1], Jouni Pulliainen[1]

[1]Space and Earth Observation Centre, Finnish Meteorological Institute, Helsinki, FI-00560, Finland
[2]Geoinformatics Research, Finnish Environment Institute, Helsinki, FI-00790, Finland
[3]Program for Environmental Information, Finnish Environment Institute, Helsinki, FI-00790, Finland

*Correspondence to*: Henna-Reetta Hannula (Henna-Reetta.Hannula@fmi.fi)

**Abstract.** We publish and describe a surface spectral reflectance data record of seasonal snow (dry, wet, shadowed), forest ground (lichen, moss) and forest canopy (spruce and pine, branches) constituting the main elements of the boreal landscape. The reflectances are measured with spectro(radio)meters covering the wavelengths from visible (VIS) to short-wave infrared (SWIR) (350 to 2500 nm). In this paper, we describe the instruments used and how the spectral observations at different scales along with the concurrent in situ reference data have been collected, processed and archived. Information on the quality of the data and factors causing uncertainty are discussed. The main experimental site is located in Sodankylä Arctic Space Centre in northern Finland (67.37° N, 26.63° E, 179 m.a.s.l) and the surrounding region. The collection includes highly controlled snow and conifer branch laboratory spectral measurements, portable field spectroradiometer observations of snow and snow-free ground at different locations and continuous mast-borne reflectance time series data of a pine forest and forest opening. In addition to the surface level spectral reflectance, data from airborne imaging spectrometer campaigns over Sodankylä boreal forest and Saariselkä fell region at selected spectral bands are included in the collection. All measurements of the data record correspond to a typical polar orbiting satellite observation event in high latitude spring season regarding their sun or illumination source (calibrated lamp) zenith angle and close to nadir instrument viewing angle. For all measurement geometries, observations are given in surface reflectance quantity corresponding to the typical representation of a satellite observation quantity to facilitate their comparison with other data sources. The openly accessible spectral reflectance data at multiple scales are suitable e.g. to climate and hydrological research and remote sensing model validation and development. To facilitate easy access to the data record the four datasets described here are deposited in a permanent data repository (http://www.zenodo.org/communities/boreal_reflectances/) (Hannula et al., 2019). Each dataset of a distinct scale has its own unique DOI (laboratory: 10.5281/zenodo.3580078 (Hannula and Heinilä, 2018a), field: 10.5281/zenodo.3580825 (Heinilä et al., 2019a), mast-borne: 10.5281/zenodo.3580096 (Hannula and Heinilä, 2018b), airborne: 10.5281/zenodo.3580451 (Heinilä, 2019a), and 10.5281/zenodo.3580419 (Heinilä, 2019b).

## 1. Introduction

High latitude regions are facing fundamental and rapid changes in climate and hydrology due to raising of the mean annual temperatures (ACIA, 2005; AMAP, 2017). The climate warming induced changes in snow cover appearance, shifting of the vegetation zones and distribution of animal species, have complex impacts on ecosystems and people (Brown and Mote, 2009; Callaghan et al., 2011). Monitoring of the seasonal snow cover of the spatially vast sub-arctic and boreal zone benefits from remote sensing for various scientific and socio-economic uses, e.g. related to the assessment of carbon balance in the boreal and sub-arctic forests (Böttcher et al., 2014; Pan et al., 2011; Pulliainen et al., 2017). Remote sensing has developed over decades into an increasingly reliable and cost effective way to estimate the decadal and annual changes in the Northern Hemisphere terrestrial snow cover (Brown and Robinson, 2011; Choi et al., 2010; Derksen and Brown, 2012; Dietz et al., 2012; Frei et al., 2012; Hori et al., 2017). The development of reliable methods to map snow extent, including the fractional snow cover (FSC), remains a challenging task especially due to the disturbing effect of forest canopy and heterogeneous land cover. Several approaches have been used to estimate the FSC from satellite imagery (Klein et al., 1998; Hall and Riggs 2007; Dozier et al., 2009; Nolin, 2010; Dietz et al., 2012; Frei et al., 2012; Metsämäki et al., 2015). These methods, such as semi-empirical reflectance model-based method SCAmod (Metsämäki et al., 2005, 2012), used for the detection of snow cover in forested areas, have benefited from accurate reference spectral measurements enabling better characterization of the model parameters (i.e. spectral endmembers). Spectral endmember refers to a 'pure' reflectance spectra of a distinct surface type such as distinct type of snow or tree species.

Field spectroscopy techniques have evolved into a widely used tool to understand the effects of the measured target on the propagation of electromagnetic radiation. This provides observations under more controlled conditions compared to e.g. measurements from satellite platforms (Aoki et al., 2000; Bänninger et al., 2008; Horton and Jamieson, 2017; Milton et al., 2009; Painter et al., 2013; Peltoniemi et al. 2005; Pirazzini et al., 2015; Tanikawa et al., 2014). In order to establish improved optical snow mapping methods for forested areas, detailed surveys of satellite scene reflectance contributors are required, as the relatively large satellite footprint may contain both fractional snow and forest cover. Additionally, snow characteristics may vary according to land cover type, e.g. between forests and open areas. Using continuous spectral signatures, i.e. from instruments with very narrow bandwidths, various land cover constituting elements can be spectrally characterized and their contribution to satellite scene reflectance then identified. In boreal landscape, reference spectroscopy measurements are valuable in defining the spectral endmembers of the reflectance, namely snow, snow-free terrain after melting and forest cover. This data can be obtained from controlled condition laboratory spectroradiometer observations, portable field spectroscopy campaigns, mast-borne spectral monitoring and aerial surveys. These approaches provide observations at different scales. Laboratory measurements can generate detailed information e.g. on spectral signature of a trunk, branch or leafs of a single tree, whereas using portable field spectroscopy, several land cover categories or shrub layer vegetation types can be spectrally characterized. Mast-borne monitoring of scene reflectance facilitates time-series production and study of the seasonal

behaviour of fractional snow and forest covered scene reflectance. Aerial surveys are useful in extending the observations to
a larger variety of landscape properties in particular during the melting season and still maintaining the advantage of high
spatial resolution.

To fully benefit from the increasing amount of spectral reflectance data records available in various archives and libraries it is
essential to ensure that the data are of consistent quality and accompanied with information on the sources of uncertainties,
such as the variations in the incoming radiation during the measurements or unideal characteristics of the measurement setup
or the used reference calibration target. However, in the case of field measurements in natural environment, it is difficult to
provide quantitative uncertainty information due to the lack of repeatability of the exact same conditions on different occasions
(even though single measurement may include several spectral acquisitions). In addition, there are limiting factors due to the
ambiguous use of reflectance terminology, measurement geometry description and variable measurement protocols (Milton et
al., 2009; Schaepman-Strub et al., 2006). To control the limiting factors the provision of metadata and sufficient documentation
on the measurement conditions and target characteristics is essential (Rasaiah et al., 2014, 2015). To produce high quality
spectral information, some guidelines for successful measurements and factors influencing the measurement output have been
reported by both the instrument manufacturers and by individual scientific projects (Goetz, 2012; Pfitzner et al., 2011).
Depending on the application, different levels of quality can be acceptable, but in general, common protocols and standardized
terminology are required for successful data sharing, fusion of different data sources and for the data comparison (Dor et al.,
2015; Milton et al., 2009). Often laborious and time consuming experimental field work, data review and quality check may
limit the resources to compile a thoroughly described metadata (Kokaly et al., 2017; Rasaiah et al., 2015). A set of metadata
parameters critical to field spectroscopy have been presented by Rasaiah et al. (2014). They include viewing geometry,
location, general target and sampling properties, illumination, instrument properties, reference standards, calibration,
hyperspectral signal properties, atmospheric conditions, and general project details. The requirements aim at such metadata
and documentation that the user is able to assess the quality level of the spectra and account for the likely variations from one
data record to another.

We collected a surface spectral reflectance data record including main components (model spectral endmembers) of boreal
seasonally snow covered landscape during spring. The collection consists of laboratory, portable field, mast-borne and selected
airborne campaign reflectance observations of snow and vegetation representing a sub-arctic site in northern boreal forest
zone. The site is located at the Sodankylä Arctic Space Centre in northern Finland (67.37° N, 26.63° E) including the
surrounding regions. The mast-borne data records are available since 2010 until 2018. Data from other platforms (portable
field and airborne systems) incorporate targeted campaign data records overlapping in time (2010–2011) with the mast-borne
observations. The data record is constructed according to the principles described above, hence enabling the utilization by a
diverse user community. Two spectrometer types were used to assemble the data record: 1) Analytical Spectral Devices (ASD)
field portable spectroradiometers covering the range from 350 nm to 2500 nm with a spectral resolution of approximately 3

nm at 700 nm and 10–12 nm between 900–2500 nm  and 2) AisaDUAL airborne imaging spectrometer covering the range from 400 nm to 2500 nm with a spectral resolution of 5 nm between 400–970 nm and 6 nm between 970–2500 nm. The data record is composed of ASD spectral signatures and AisaDUAL image mosaics in selected wavelength bands (555 nm, 645 nm, 859 nm and 1640 nm). In addition to the spectral reflectances, the associated metadata describe the utilized instrumentation, measurement protocol, target properties, information about impurities in snow and measurement/environment conditions, such as weather and illumination. Here we first describe the Sodankylä main experimental site and give a concise overview of the measurement systems and observed targets. Then, we define the provided reflectance quantity of the data record. From then on, the measurement systems and measured targets of each scale/platform are introduced in detail and examples of each scale of the data record are given. Finally, discussion of possible sources of error and uncertainty is accompanied with conclusions and recommendations for data utilization.

## 2.  Study area and spectral measurements

### 2.1 Sodankylä site in Northern Finland

In this section, we describe the Finnish Meteorological Institute's Sodankylä Arctic Space Centre (FMI-ARC) experimental site characteristics for data collection. Besides the four different spatial scale datasets (described in the following Sections 2.2 and 3) from the FMI-ARC area, data is also presented from one aerial survey over Saariselkä, around 120 km north of Sodankylä. Saariselkä is a fell (arctic hill) region that has a timberline at an altitude of 400 m above sea level. The treeless altitudes represent fell tundra (Virtanen et al., 2016). Also, some individual field spectral samples from boreal forest area in Nuuksio, Espoo, southern Finland, are included in the collection (Fig. 1). This additional data was collected to capture more observations from late melting conditions of the snowpack.

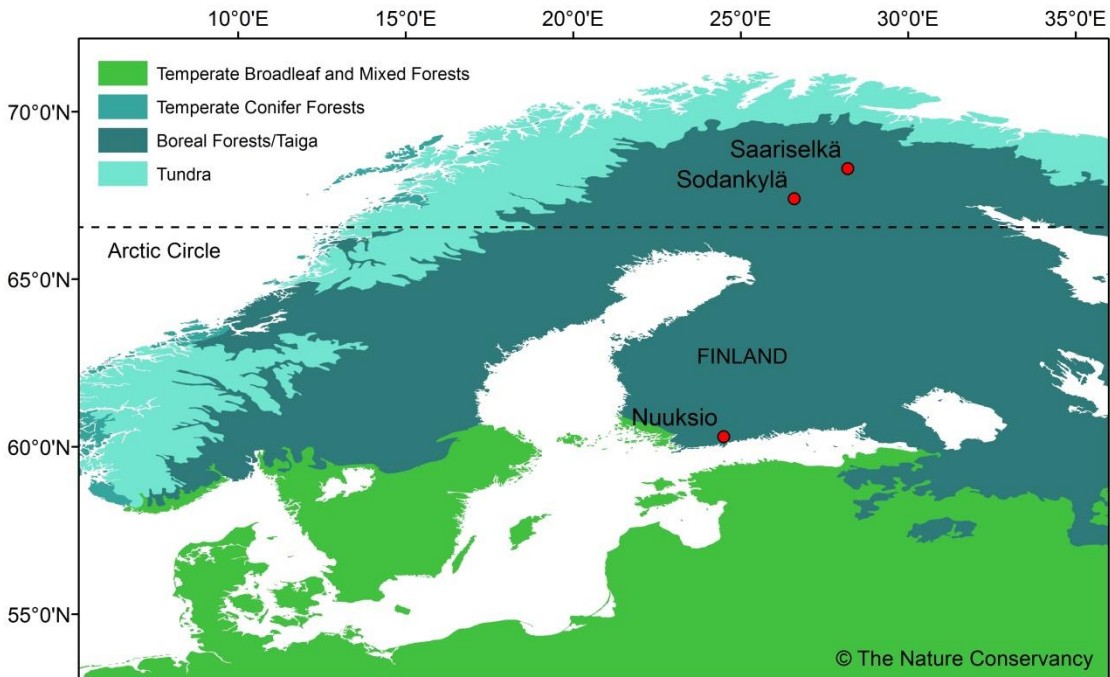

**Figure 1. The location of Sodankylä Arctic Space Centre (FMI-ARC), where most of the data records have been measured. In addition, one aerial survey was conducted in Saariselkä fell region, north from the FMI-ARC, and some individual field spectra were measured in Nuuksio, Espoo, southern Finland. Distribution of boreal, temperate conifer, temperate broadleaf and mixed forests, and tundra by the Nature Conservancy (Olson and Dinerstein, 2002, maps.tnc.org).**

The Sodankylä station, situated above the Arctic Circle in northern Finland (67.37° N, 26.63° E, 179 m.a.s.l.), provides an ideal location for environmental and atmospheric research of the boreal and sub-arctic region. In fact, Sodankylä is one of the primary stations of the WMO Global Atmosphere Watch (GAW), WMO Global Cryosphere Watch (GCW) and Integrated Carbon Observation System (ICOS) networks. Measurements at the Sodankylä station date back for more than 100 years (Kangas et al., 2016). At the Sodankylä station and its vicinity, there are automated, as well as, manual in situ data collection in a variety of different ecosystems and landscapes, in addition to several permanent platforms for satellite reference instruments.

The Sodankylä region is a globally representative example of the boreal forest biome, which encompasses the largest continuous land ecosystem on the Planet (ACIA, 2005). Seasonal snow cover is a characteristic feature of the boreal forest zone affecting strongly the functions of the ecosystem, water cycle and surface-atmosphere interaction. Sodankylä has a sub-arctic climate due to the warming effect of the Gulf Stream (Kangas et al., 2016) and the area represents taiga snow.

Characteristics for Sodankylä are extreme seasonal temperature variations as well as long and cold winters with snow season from October until May. Sodankylä area represents taiga snow and from 1981 till 2010, the maximum snow depth of approximately 80 cm occurred in late March (Pirinen et. al., 2012). The changing seasonal snow cover affects the boreal forest carbon uptake and storage and hydrological cycle that are important features of the boreal ecology also in Sodankylä region (Pan et al., 2011). The landscape around the Sodankylä station is relatively flat, with isolated fells reaching up to 500 m. The landscape consists of sparse pine dominated coniferous forests and open areas on mineral soil as well as open peat bogs (Leppänen et al., 2016).

## 2.2. Spectrometer measurement geometry, calibration and reflectance quantities

All spectral observations here, regardless of their measurement scale, correspond to a typical polar orbiting satellite measurement in high latitude spring season with respect to their sun or illumination source (calibrated lamp) zenith angle and close to nadir instrument viewing angle. Fig. 2 illustrates the radiance (unit of measure: W m-2 sr-1 nm-1) measurements conducted using different platforms to provide reflectance data for satellite observation analysis. Satellites observe the radiation intensity reflected from the Earth's surface, which is calibrated into a physical quantity, such as top of atmosphere (TOA) reflectance that can be converted to surface reflectance using atmospheric correction methods. Here, the spectra are calibrated to surface reflectance, by using a white reference panel with a known reflectance spectrum approximating a Lambertian surface; the incoming radiation is determined by measuring the radiance from a white Spectralon panel (12.7 cm, SRT-99, Labsphere inc., USA). Spectralon is made of packed sintered polytetrafluoroethylene (PTFE) powder which is highly reflective and exhibits nearly Lambertian behaviour from ultra-violet to near-infrared region. PTFE is chemically stable and hydrophobic (Springsteen, 1999). The calibration procedure for each platform was conducted as follows.

In laboratory, the Spectralon radiance was measured both before and after each pine/spruce branch sample and the sample radiances were converted to absolute reflectance by dividing with the Spectralon radiance and multiplying with the reference panel calibration data (from the manufacturer). In the case of snow measurements in the laboratory, the Spectralon radiance was measured in the beginning and in the end of the measurements of samples of the specific snow type and all the Spectralon radiances for each measurement day were then averaged. In field measurements, the Spectralon was measured before each (snow or lichen/moss) sample and repeated when necessary, e.g. if the illumination conditions changed during one measurement event.

Mast-based (forest opening and pine forest) target radiances were converted to reflectance by using a Spectralon radiance measurement obtained before each target observation (the Spectralon is pushed under the measurement head automatically). The instrument is taken down from the mast for the cold and lightless mid-winter. During this time, dark laboratory tests are conducted to reveal any substantial changes in the instrument response or possible degradation of the Spectralon panel due to impurities or exposure to UV (ultraviolet) radiation. The changes in the reference panel reflectance are tested by measuring

the Spectralon against a similar panel without exposure to any external stresses. In most cases, these measurements are executed before and after cleaning the panel (pressure air or sanding under running water), the former status of the Spectralon being valid for mast measurements before, and latter for measurements after the laboratory tests. The observed mast scene absolute reflectance values ($R_{scene}$) at wavelength $\lambda$ are then corrected based on the laboratory results as follows:

$$R_{scene}(\lambda) = R_{cal}(\lambda) * \frac{L_{ref}(\lambda)}{L_{cal}(\lambda)} * \frac{L_{scene}(\lambda)}{L_{Ref}(\lambda)} \tag{1}$$

where $R_{cal}$ is the Spectralon panel calibration data from the manufacturer, $L_{ref}$ is the Spectralon panel radiance for the mast reference and $L_{cal}$ Spectralon panel radiance for the laboratory standard measured together in a dark laboratory. $L_{scene}$ is the radiance of the target scene and $L_{Ref}$ the radiance of the Spectralon panel measured at the mast.

Airborne radiances measured in March 2010 were converted to reflectances by vicarious calibration. Airborne radiances were compared with the concurrent mast-borne radiances from the forest site and calibration coefficients were determined for AISA data by using least squares fitting technique. To obtain reflectances the concurrent calibrated mast-borne Spectralon radiances were utilized (Heinilä et al. 2014). In 2011, the airborne reflectance level was obtained by applying a real-time fibre optic downwelling irradiance sensor (FODIS).

The reflectance quantity of all observations discussed here, corresponding to atmospherically corrected estimate of surface reflectance from satellite data, is

$$R = \pi \frac{L_{obs}}{E_0 \cos(\theta_i)} \tag{2}$$

where

$R$ is the surface reflectance factor

$\pi$ is a scaling factor related to Lambertian surface

$L_{obs}$ is the instrument-observed radiance

$E_0 \cos(\theta_i)$ is the incoming radiance projected to the surface, the Sun zenith angle of incident radiation is $\theta_i$.

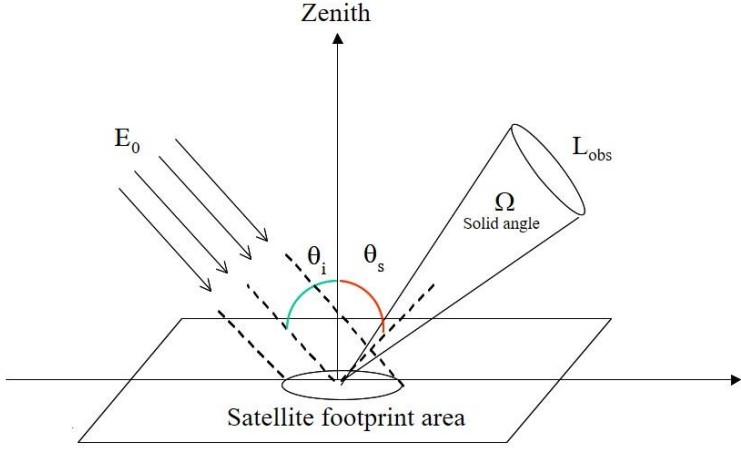


**Figure 2. General concept of a satellite or ground based remote sensing measurement of reflected radiance. $\theta_i$ is the incidence angle of incoming irradiance and $\theta_s$ is the instrument's viewing angle. The incoming Sun irradiance $E_0$ is projected to the Earth's surface (instrument's footprint area) with the magnitude $E_0 \cos(\theta_i)$. The instrument measures the reflected radiance $L_{obs}$ within its viewing angle (i.e. the radiant flux per unit solid angle, $\Omega_{solid\ angle}$). The ratio between the reflected and incoming radiation provides**

**reflectance according to Equation (2). The azimuth angle is omitted for simplicity. The figure is adapted from Salminen (2017).**

The reflectance data given here by Equation (2) specifically represent the conceptual quantity of directional-hemispherical reflectance (Shaepman-Strub et al., 2006). This is the case, since the calibration is carried out by a white reference panel approximating a Lambertian surface, and the incoming irradiance is predominantly or totally originating from one (narrow)

direction of the illumination source (calibrated lamp or the Sun). Since the calibration is a comparison against a Lambertian surface, the recorded reflectance can show values above one. Additionally, some observations are obtained under diffuse illumination conditions (full cloud cover providing close to hemispheric isotropic illumination), but using the same calibration procedure with a white reference panel.

**2.3. Measurement systems and collected data**

In this section we give a short overview of the measurement setups and platforms. In Chapter 3 below, the conditions and processing steps for the data collection are described in more detail. The four platforms included are laboratory, portable field, mast-borne and airborne setup (Fig. 3). The instrument utilized for the first three platforms is Field Spec Pro JR Spectroradiometer by ASD (Boulder, Co, USA). The laboratory and field measurements were carried out with the same Field

Spec Pro JR unit, whereas the mast-borne instrument is a fixed installation. The AisaDUAL airborne imaging spectrometer by SPECIM corp. (Oulu, Finland) was used on the airborne platform. The technical specifications and details of the setups of ASD and AisaDUAL are described in Table 1. An overview of the measurement systems and targets along with the Digital

Object Identifier (DOI) for each dataset are given in Table 2. The overlap of the different measurements conducted over different platforms are presented in Fig. 4. In brief, spectra of snow, pine branches and spruce branches were measured by using the laboratory setup (Fig. 5), whereas snow on ground and snow-free ground spectra were obtained with the portable setup (Fig. 7). In both cases the footprint of a single measurement was small in the order of 4–20 cm (diameter) depending on the measurement optics and the distance between the sensor and the target. Mast-borne forest and forest opening spectra were collected during winter and spring-melt periods with a footprint size of about 14 m in diameter. Selected spectral bands from the airborne AisaDUAL measurements from snow and snow-free ground surveys were added to the published collection. The investigated targets are somewhat different between the four scales, but they are the components of the same investigated boreal landscape.

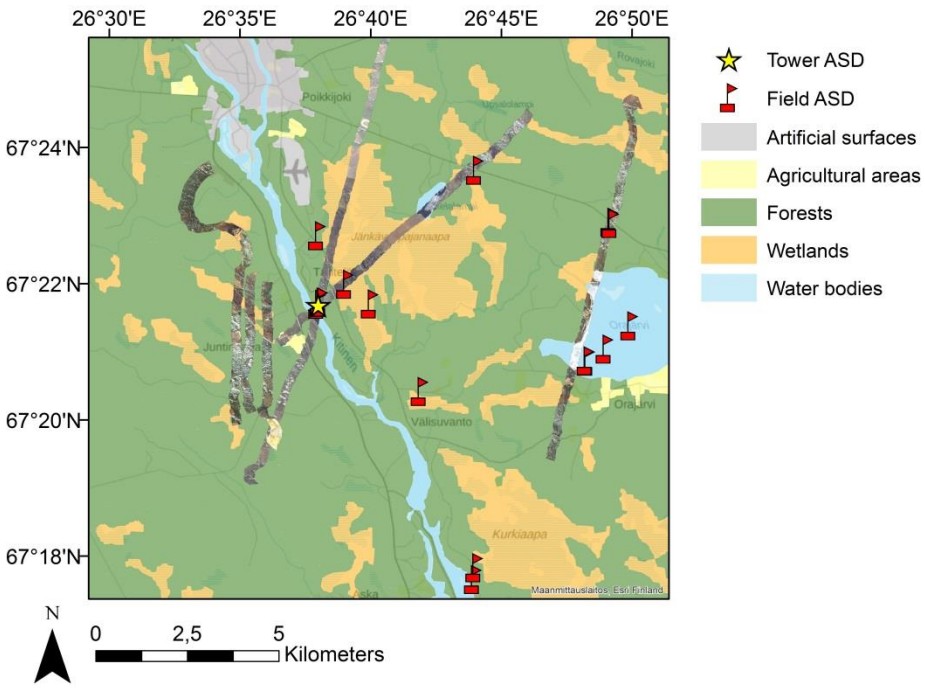

**Figure 3. AisaDUAL flight lines measured in 2010 and 2011 and the measurement points of the portable field and the mast-borne measurements at the FMI-ARC main site. In the background Corine Land Cover 2018 classification by Copernicus Programme and a basemap © National Land Survey of Finland, Esri Finland 12/2018.**

**Table 1. Technical details of instruments and different installation platforms.**

| | ASD Field Spec Pro JR | | | AisaDUAL | |
| --- | --- | --- | --- | --- | --- |
| | Laboratory (SYKE #6424) | Portable Field (SYKE #6424) | Mast-borne (FMI #6484) | AisaEAGLE sensor | AisaHAWK sensor |
| Detector | silicon photo diode array detector (350–1050 nm), indium gallium arsenide photo-diode detectors (900–1850 and 1700–2500 nm) | | | CCD 12 bits | MCT 14 bits |
| Wavelength region (nm) | 350–2500 nm | | | 400–970 nm | 970–2500 nm |
| Spectral resolution | 3 nm (350-1000 nm) 10–12 nm (1000-2500 nm) | | | 5 nm | 6 nm |
| Spectral bands | ~367 | | | 359 | |
| Measurement head | Fiber optic / Foreoptic | Fiber optic | Fiber optic | Foreoptic | |
| Field of view (FOV) | 25° / 8° | 25° | 25° | 17° | |
| View zenith angle | 0° / 0° | 0° | 11° | 0° | |
| View azimuth angle | 0° / 0° | 0° | 109° (forest) 267° (forest opening) | 0° | |
| Fiber optic head distance to target/ Flight altitude | 25 cm | 45 cm | 30 m (ground) | 800 m | |
| Footprint/ Spatial resolution | Ø 11 cm / Ø 3.5 cm | Ø 20 cm | Ø 13.7 m | 80 cm x 80 cm | |
| Swath | n/a | n/a | n/a | 240 m | |

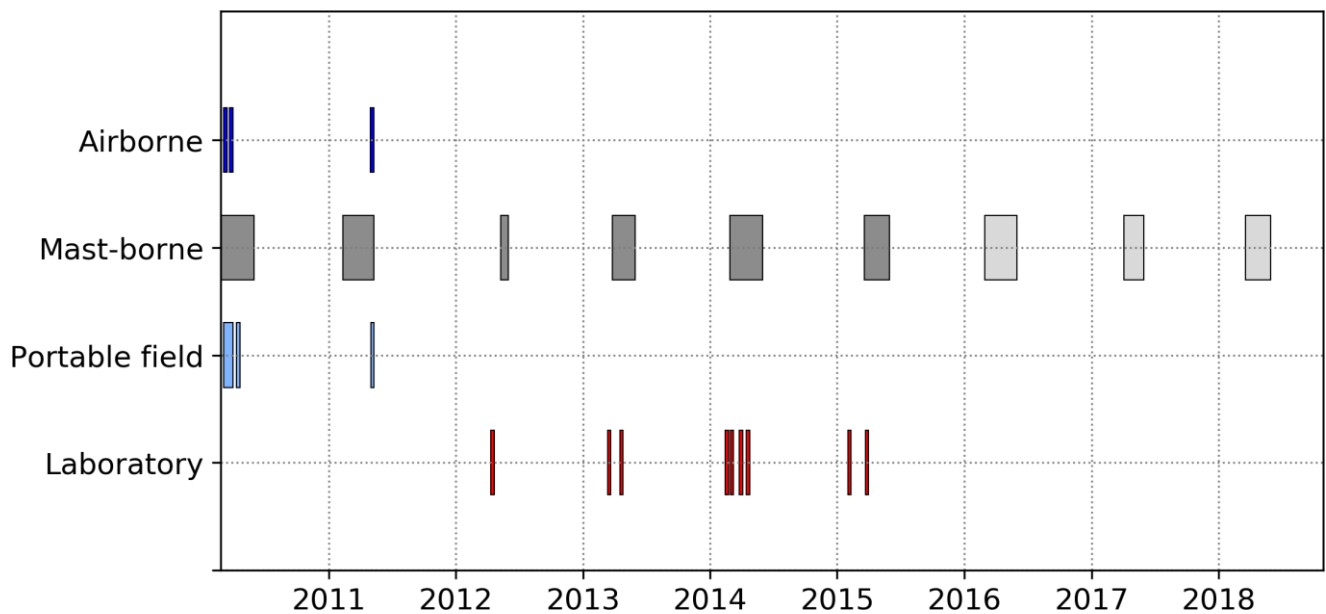

**Figure 4. Overlap of the different measurements conducted over different platforms. For the mast-borne platform, the observations first cover spectral range of 350–2500 nm and range of 350–1000 nm from 2016 onwards.**

**Table 2. The data record measurement dates and targets. The total number of separate measurements per target type as well as the number of spectra averaged (consequent spectral acquisitions collected at 1 second interval) for each individual target measurement are presented. The accompanied reference measurements are described in chapter 3.5. In the last column the Digital Object Identifier (DOI) for each spectral dataset is given.**

| Data record | Dates | Target characterization | No. of separate measurements /samples per target type (see target characterization) | No. of spectra averaged per individual reflectance measurement | FOV (°) | Reference measurements | DOI |
|---|---|---|---|---|---|---|---|
| | | | | | | | |

| Laboratory | 12.04.2012 | Pine branches | 15[*] | 20 | 25 | | 10.5281/zenodo.3580078 |
|---|---|---|---|---|---|---|---|
| | 12.04.2012 | Spruce branches | 15[*] | 20 | 25 | | |
| | 21.03.2013 | Dry snow in sun | 15 | 30 | 25 | In situ snow | |
| | 21.03.2013 | Dry snow in shadow | 9 | 30 | 25 | properties, | |
| | 18.04.2013 | Wet pure snow | 25 | 30 | 25 | weather | |
| | 18.04.2013 | Wet snow with littered surface | 25 | 30 | 25 | conditions, | |
| | 20.02.2014 | Dry snow | 12 | 20 | 8 | photographs of | |
| | 27.02.2014 | Dry snow | 15 | 20 | 8 | the wet snow | |
| | 27.03.2014 | Dry snow with moist surface | 12 | 20 | 8 | samples with | |
| | 16.04.2014 | Wet pure snow | 15 | 20 | 8 | littered surface | |
| | 02.02.2015 | Dry snow | 15 | 10 | 8 | | |
| | 23.03.2015 | Dry snow | 15 | 10 | 8 | | |
| **Portable Field** | 16.-22.03.2010 | Sparse pine forest, snow (dry) | 11 | 30 | 25 | Location | 10.5281/zenodo.3580825 |
| | | Sparse pine forest, snow (dry), branch shadow | 5 | | | characteristics, | |
| | | Sparse pine forest, snow (dry), trunk shadow | 1 | | | in situ snow | |
| | | Open mire, snow (dry) | 9 | | | properties, | |
| | | Pine forest, snow (dry) | 11 | | | weather | |
| | | Snow covered lake ice, snow (dry) | 19 | | | conditions | |
| | 12.04.2010[**] | Grassland/ open field, snow (wet) | 16 | | | | |
| | | Grassland/open field, snow free, open ground/soil | 3 | | | | |
| | 05.05.2011 | Sparse pine forest, snow(wet) | 3 | | | | |
| | | Sparse pine forest, snow free, lichen | 1 | | | | |
| | | Pine forest, snow (dry) | 2 | | | | |
| | | Pine forest, snow (wet), branch shadow | 4 | | | | |
| | | Pine forest, snow (wet), trunk shadow | 1 | | | | |
| | | Pine forest, snow free, moss | 1 | | | | |

| Mast-borne spectro-radiometer | 25.02-31.05.2010 | Pine forest canopy | 523 | 1 | 25 | Digital images, wind in gust at 22 m (m/s), cloudiness (octas), air temperature at 2 m (C°) from an automatic weather station, Δ-value in 2013–2018 | 10.5281/zenodo.3580096 |
|---|---|---|---|---|---|---|---|
| | 11.02-11.05.2011 | Pine forest canopy/ forest opening | 179/ 144 | | | | |
| | 09.05-31.05.2012 | Pine forest canopy/ forest opening | 30/ 31 | | | | |
| | 26.03-31.05.2013 | Pine forest canopy/ forest opening | 751/ 710 | | | | |
| | 27.02-31.05.2014 | Pine forest canopy | 848 | | | | |
| | 20.03-31.05.2015 | Pine forest canopy | 680 | | | | |
| | 29.02-31.05.2016*** | Pine forest canopy | 1075 | | | | |
| | 04.04-31.05.2017*** | Pine forest canopy | 644 | | | | |
| | 20.03-31.05.2018*** | Pine forest canopy | 902 | | | | |
| **Airborne spectrometer** | **Measurement days/ area** | **Target characterization** | **Extracted bands** | **Stripe width/ Resolution** | **FOV (°)** | **Reference measurements** | **DOI** |
| | 18.03.2010/ Sodankylä | Dry snow cover, snow-free forest canopy | 555 nm, 645 nm, 858.5 nm, 1640 nm | 240 m/ 10 x 10 m$^2$ | 17 | Portable field measurements and the accompanied reference measurements | 2010: 10.5281/zenodo.3580451 2011: 10.5281/zenodo.3580419 |
| | 21.03.2010/ Sodankylä | Dry snow cover, new snow, snow-on-canopy | | | | | |
| | 05.05.2011/ Sodankylä and Saariselkä | Thin melting snow cover, snow-free patches, snow-free forest canopy | | | | | |

* Box with pine/spruce branches was shifted 15 times per measurement geometry

** Measured in grassland surroundings in Southern-Finland

*** Data available only between 350–1000 nm

## 3. Dataset description

### 3.1 Laboratory spectroradiometer measurements of snow, pine and spruce

Snow, pine and spruce branch reflectance were measured in laboratory conditions to define the endmember reflectances used in the optical remote sensing of snow (Metsämäki et al., 2005, 2012). The experiments were carried out with the same ASD Field Spec Pro Jr Spectroradiometer that was also used in the field measurements. Reflectances of pine and spruce branches were measured in April 2012. The laboratory measurements of snow reflectances were conducted in springs of 2013, 2014 and 2015 (Hannula and Pulliainen, 2019). The snow measurements were done for different snow types. The properties of snow were also measured in situ (Table 3 in chapter 3.5).

The pine and spruce branches were collected on the experiment day and placed inside a fridge until they were measured. Two black boxes were filled with the branches, one with pine branches and one with spruce branches. Bare fiber optic was used as a measurement head and spectra were collected at nadir view angle. The laboratory measurement setup for the spruce spectra acquisition is illustrated in Fig. 5a. A calibrated Tungsten halogen lamp was used as a light source with a light zenith angle of $\theta$=55°. The lamp current was stabilized at 8 A with accuracy of 0.01 % leading to accuracy of lamp irradiance of +/- 0.1 %. In each case, a mean from 20 individual spectra, measured at one second intervals, was calculated. For both, pine and spruce, the measurements were repeated 15 times shifting the box randomly to different position between the measurements. The fiber optic head distance to the target (Table 1) was defined from the uppermost limit of the pine/spruce branches.

For the snow sample collection, an aluminium sampler of the size of 35 cm x 35 cm x 23 cm, painted inside with matte black colour was used. After removal, each sample was placed inside a black insulated box, carried inside the laboratory and measured immediately (Fig. 5b). A mean of 10, 20 or 30 individual spectrum acquisitions (depending on snow type), collected at one second interval, from each snow sample were measured. The measured snow type conditions represented wet melting snow (N=2), wet melting snow affected by forest litter inclusions (N=1), dry snow with moist surface (N=1) and dry snow (N=6). The illumination zenith angle was $\theta$=55°. Bare fiber optic was used as a measurement head in 2013 and spectra were collected at nadir view angle. In 2014 and 2015 an eight degree foreoptic was used. The number of snow samples collected and measured from each snow type varied between 9 and 25, in total (Table 2).

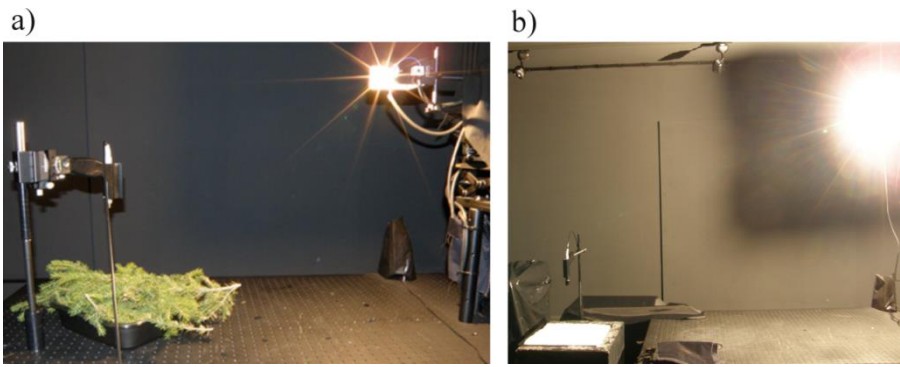

**Figure 5. Measurement set-up for a) spruce branches and b) for snow samples.**

(a)

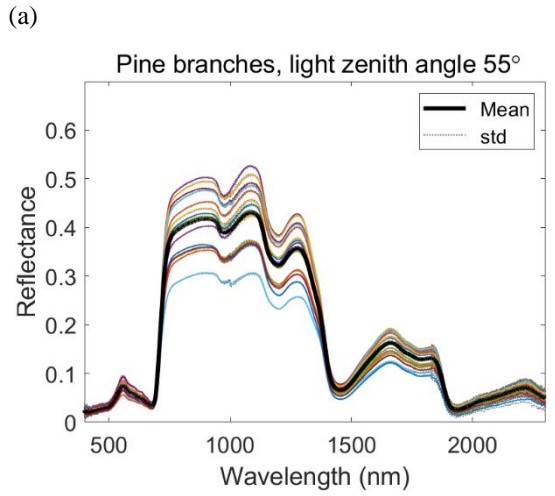

(b)

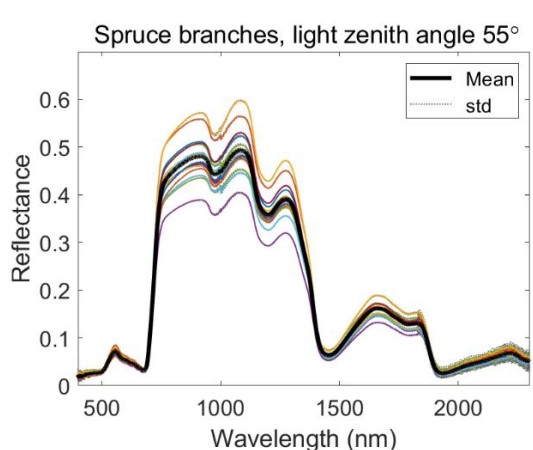

(c)

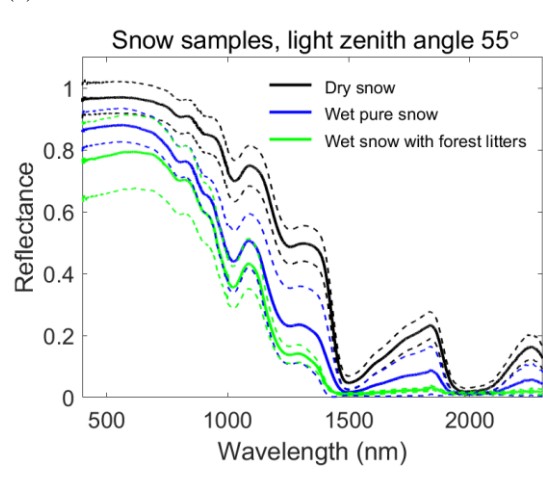

**Figure 6. Reflectance spectra for a) pine branches b) spruce branches and c) three types of snow measured in dark laboratory conditions +/- standard deviation. In a) and b) individual reflectance spectra for pine and spruce branches measured by shifting the sample box are presented and the standard deviation is the deviation between consequent measurement acquisitions (n=20) averaged for each individual spectrum. In c) the standard deviation is the deviation between different snow samples collected and measured from the same snow type investigated.**

Examples of the mean spectra measured in laboratory for pine and spruce branches and dry and wet snow types are shown in
Fig. 6. The spectra in Fig. 6a and 6b show the variation in pine/spruce branch reflectance when the sample box was shifted
under the measurement head. The standard deviation, defined from the consequent measurement acquisitions collected and
averaged for one measurement spectrum, is so small that they are very difficult to distinguish from the plot. In Fig. 6c the
standard deviation shows the deviation between different snow samples separated and measured from the same investigated
snow type. Although, the experiment setup does not simulate the true conditions accurately, in laboratory the measurement
surroundings can be controlled. Practically removing the effects of changing illumination conditions and diffuse light, it is
possible to evaluate the spectral characteristics of the targets. The main findings of the laboratory experiments are the mean
and standard deviation of reflectance for dry and wet snow types as well as for boreal pine and spruce branches. These provide
reference information to be utilized in the characterization of the spectral endmembers of the remote sensing models.

## 3.2 Portable spectroradiometer measurements of snow and snow-free ground

The reflectance spectra from snow and the ground underneath the snow cover were measured with the ASD Field Spec Pro
JR., which was also used in the laboratory measurements. Timing of the measurement campaigns was aimed to be both during
the cold season and during the melting period, when patches of open ground appear in the snow surface and when the snow
properties have higher variation. The measurement targets were characterized with location, landscape characteristics, weather
conditions, namely air temperature and cloud cover, and in situ snow properties. Measurements were carried out in springs of
2010 and 2011 for Sodankylä station and in spring 2010 on the grassland site in Southern Finland. The data from Sodankylä
was collected parallel to airborne campaigns with the AisaDUAL imaging spectrometer.

For field measurements, the spectroradiometer was placed in polypropylene case with soft interior padding to protect the
instrument during transport. Additionally, an external battery and a laptop were connected to the measurement unit. The
measurement head was mounted on a camera tripod with an arm that can extend the measurement head around 40 cm from the
centre of the tripod base (Fig. 7). The tripod was placed the arm extending towards the sun. A second tripod with a bubble
level was used to place the Spectralon panel horizontally under the measurement head for the reference measurements. At each
measurement location the coordinates and general conditions were logged and the incoming full sky irradiance was measured.
For the reflectance measurements, first the Spectralon reflectance standard was measured and then the reflectance spectrum of
the target and e.g. in situ measurements of snow were carried out (Table 3). The distance between the tip of the measurement
head and the target (snow surface/ground) was approximately 45 cm and the associated full sky irradiance, measured with the
remote cosine receptor (RCR) was also measured from this height. With 25 degree field of view (FOV) of the optical fiber
head, a footprint diameter of 20 cm on the ground/target was observed.

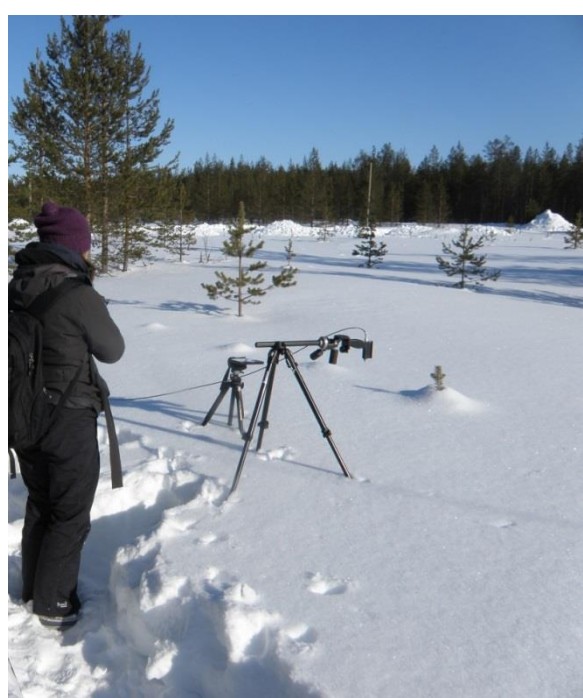

**Figure 7. Measurement set-up for field measurements with the ASD Field Spec Pro JR spectroradiometer. During the measurement, the operator moved further off and squatted to minimize the effect on the measurements.**

Measurements were carried out in forests (N =40), wetlands (N=9), grasslands (N=19) and on lake ice (N=19). The first campaign took place in 16−22 March 2010 when the area was characterized by full dry snow cover. The mean air temperature during the measurements was -5 °C. Therefore, the campaign was carried out in dominantly dry snow conditions. Snow depth for the measured sites varied, depending on the land cover type, between 12 and 87 cm. Largest snow depth was observed in a forest site and smallest on lake ice. In contrast to the first campaign, during the second campaign on 5 May 2011 the snow cover was patchy (50-60% snow patchiness) and wet. Due to varying illumination conditions during these measurements only selected spectral measurements in forest with good data quality were retained in the data record. The mean air temperature was 10.1 °C and snow depth ranged between 0−28 cm. In addition to campaign data from Sodankylä, reflectance measurements were carried out during the melting period along a transect with varying snow depth (0−26 cm) on a grassland site in Espoo, Southern Finland, on 14 April 2010. At this time the snow patchiness ranged between 50−70% and snow was very wet.

In the portable field measurements of reflectance spectrum from snow and the ground underneath the snow cover, the goal was to get better understanding of the variation of the snow reflectance under different snow conditions (e.g. with different snow depths). In Fig. 8 field snow reflectance observations in clear sky conditions in direct light and in shadow for dry and melting snow and for melting snow with different total snow depths are presented. Observed reflectances drop with increased water content, impurities and larger grain size in melting snow. The detection of snow cover in forested areas from

optical satellites is also influenced by the shadowing of the ground by trees. The shadows decrease the reflectance considerably. It should be noted that the measurements are of apparent reflectance, i.e. reflectance measured at the earth observation

instrument and therefore related to the full sky irradiance (Salminen et al., 2009). Depth of the snow pack becomes an important factor when snow cover is at melting stage and light is passed through to the ground (Salminen et al., 2009). Noise at the water absorption band, characteristic for field measurements, is seen at 1900 nm, where the signal to noise ratio is inadequate for meaningful observations. Utilising field observations, it is possible to study the effect of both the observation geometry and the target properties on the observed reflectance spectra, although controlling the measurement geometry is difficult.

(a)                                                                                              (b)

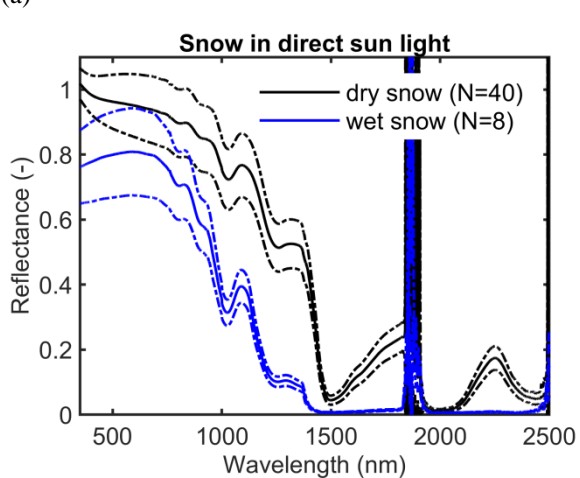    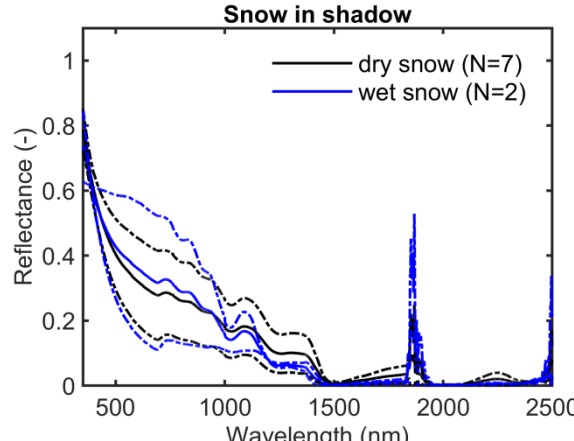

(c)

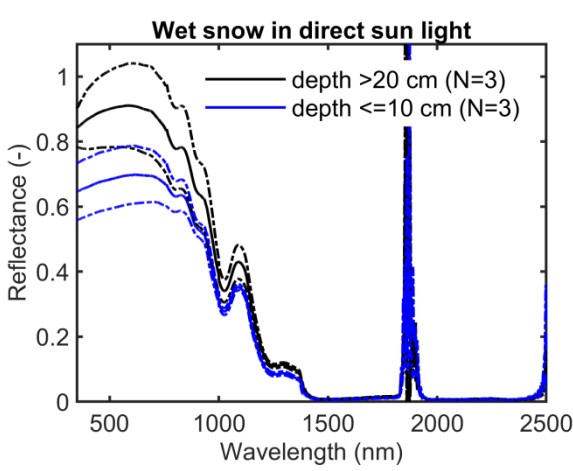

**Figure 8. Field reflectance measurements of snow. Cloudiness for all measurements was less than 3 octas. Dry snow (a–b) and wet snow (a–c) correspond to class 1 and classes 3–4 (wet and very wet snow) according to Fierz et al. 2009, respectively. Figures display the mean reflectance (solid line) and standard deviation (dashed line).**

### 3.3 Mast-borne spectroradiometer measurements of pine forest and forest opening

ASD Field Spec Pro Spectroradiometer was installed on a 33-meter-high mast in the intensive observation area (IOA) of the FMI-ARC for the optical remote sensing validation studies (project NorSEN, Nordkalotten Satellite Evaluation co-operation Network). The mast observations allow the evaluation of at-satellite reflectances in the spatial scale of the satellite image pixels. The dataset covers the spring time periods between 2010 and 2018. In 2010–2012 measurements were collected by an operator at hours 10, 12 and 14 UTC (Coordinated Universal Time) during clear sky or full cloud cover conditions. Additionally, measurements were made more frequently during specific measurement campaigns. The system was automatized during summer 2012 and after that spectral measurements have been collected yearly from February to November every 30 minutes from 6 till 15 UTC. As the climatic environment during the snow season is challenging, the measurement pole was fixed over the forest target on 26 of August 2013 due to frequent problems with the turning motor. After 29 September 2015 data is only available between 350–1000 nm because of breaking up of a non-replaceable part of the instrument (Table 2).

The instrument was placed inside a weather resistant box for protection. The ASD standard fiber optic cable was replaced by a longer 5 meter cable at the manufacturer, to enable mounting of the measurement head at the end of a turning pole. The measurement head is a bare fiber optic with FOV of 25 degrees giving a footprint of around 14 m in diameter (185 m$^2$). This enabled measurements from two separate locations from a sparse pine forest with a median tree height of 11 m (Niemi et al. 2012) (azimuth 109°) and from a forest opening (azimuth 270°) (Fig. 8). The forest opening measurement area is mainly covered by lichen but there are also some patches of moss (Sukuvaara et al., 2007).

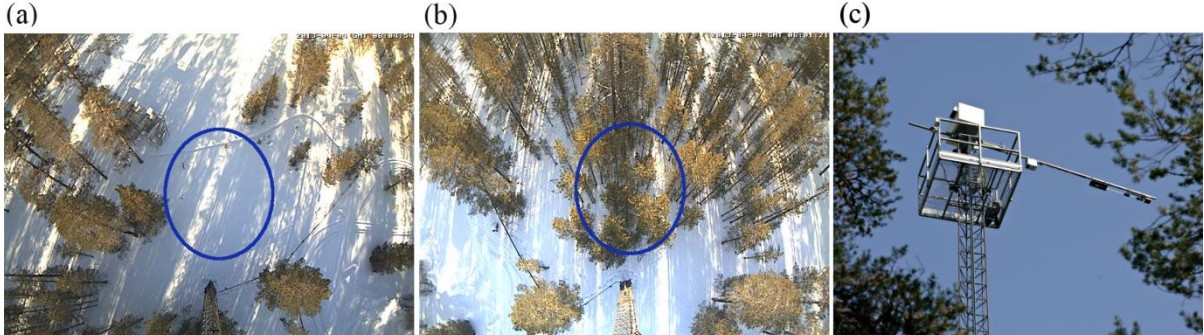

**Figure 9. Mast-spectroradiometer measurement areas of (a) forest opening, (b) sparse pine forest and (c) photograph of the mast top structure.**

The fiber optic head is tilted 11 degrees away from the mast. A calibrated white reference panel (Spectralon) is mounted under a small shelter below the turning pole and is pushed under the measurement head and measured before each target

measurement. A wide angle camera is used to image the measurement area at the time of each spectral acquisition for the
description of the measurement target characteristics.

One individual spectrum represents one instant measurement acquisition. During the automatization process threshold values
were set to avoid collection of poor data. No measurements are executed during rain or snow events, high winds (gust > 8 m/s)
or low air temperatures (< -20°C). To avoid measurements where the scene and reference spectra are collected in a considerably
different illumination conditions, the instrument collects spectra for 10 seconds before each measurement. Integrals of radiance
over wavelength are calculated and the ratio of the largest difference and the mean is returned and saved. The measurements
are collected based on a set illumination change threshold value (indicated by Δ), currently set to 10 percent. This allows
further selection of spectra based on stricter illumination standards.

The mast-borne reflectance spectra for two measurement areas, sparse pine forest and forest opening during spring 2013 were
resampled to correspond to MODIS (Moderate Resolution Imaging Spectroradiometer) band 4 (545–565 nm), essential

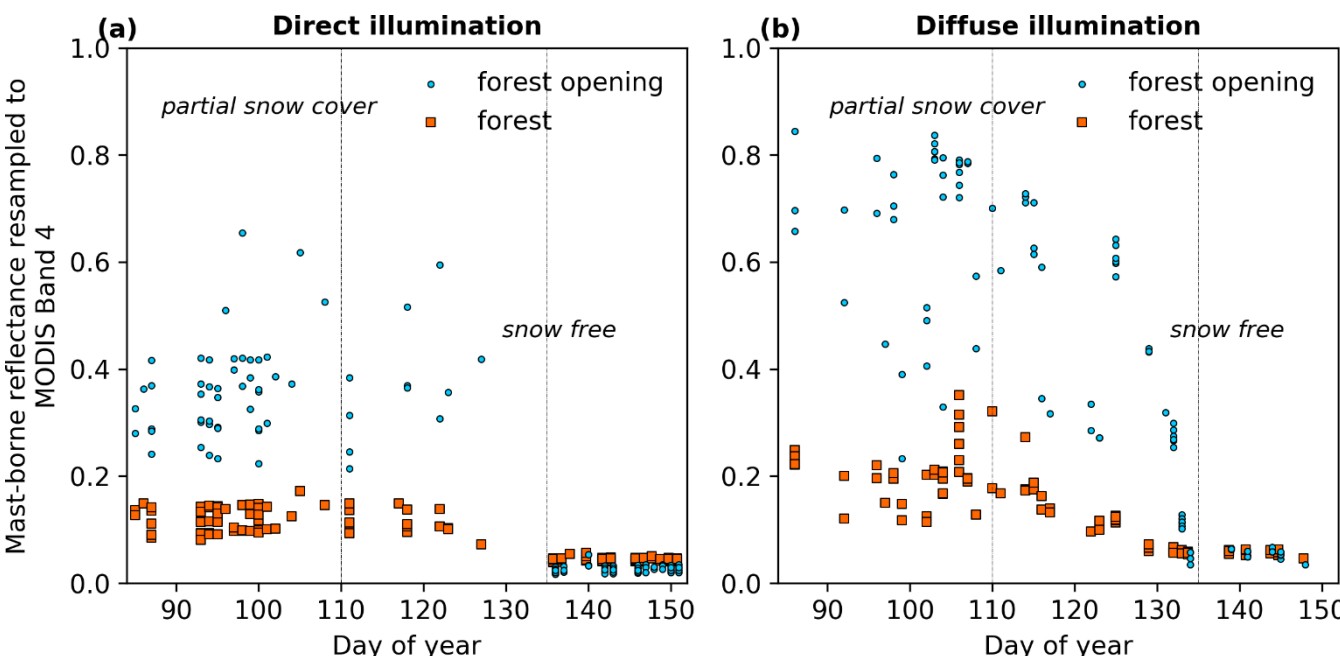

**Figure 10. Mast-spectroradiometer observations from both forest and forest opening during the spring of 2013 resampled to
correspond MODIS band 4 (545–565 nm) reflectance in (a) direct (cloudiness 0–2/8) and in (b) diffuse (cloudiness 7–8/8) illumination**
**conditions.**

for snow mapping from satellites (Fig. 10). In the resampling the corresponding wavelengths from the mast-borne spectra
were chosen and weighted averages calculated by using the relative spectral response function (RSR) provided by the data

provider. The time-series describes both the diurnal and within-season changes in reflectance. Measurements for clear sky and diffuse illumination conditions were separated. The observed values for forest opening are high compared to forest until the end of the snow season. The forest opening scene during the full snow cover is composed of snow field only whereas in the pine forest area the reflectance is dominated by the forest canopy. The casting shadows from the surrounding trees increase the reflectance variability especially for the forest opening. Considerable diurnal variation in snow reflectance during the snow cover period is also seen in diffuse illumination conditions (Fig. 10b). With automatic measurements the number of observations can be increased.

### 3.4 Airborne spectrometer survey of snow and snow-free ground

Two airborne spectral imaging campaigns were organized in Finnish Lapland. The purpose was to investigate the effect of forest canopy on optical remote sensing signals from snow-covered surfaces. The first campaign was organized in Sodankylä on 18 and 21 March 2010 and the second in Sodankylä and in Saariselkä on 5 May 2011. In both campaigns airborne hyperspectral data was acquired with the AisaDUAL imaging spectrometer manufactured by Spectral Imaging Ltd (SPECIM). The technical details of AisaDUAL sensors are presented in Table 1. The data record contains 10 meter resolution reflectance mosaics of the flight lines for the bands 555 nm, 645 nm, 858.5 nm and 1640 nm for all measurement days and for both (Sodankylä and Saariselkä) study sites (Table 2).

During the first campaign, in 2010, the ground was covered by thick (> 70 cm) dry snow layer. On 18 March, the trees were snow-free and the last clear snow fall event, based on the observations from the Sodankylä weather station, was on 3 March, therefore the snow on the ground was several days old, while on 21 March, the trees were snow-covered and the snow on trees and on ground was newly fallen. All measurements were carried out in direct illumination conditions (0/8 to 2/8 cloud cover). The AisaDUAL spectrometer was installed in a helicopter (Fig. 11). To convert the measured airborne radiances to reflectances, the concurrent mast-borne Spectralon radiances were utilized to determine the incoming radiation at the particular wavelength. To get the same reflectance level with the mast-borne observations, calibration coefficients were determined and utilized for the AISA data (Heinilä et al. 2014).

During the second campaign, in 2011, the spring snow melting was ongoing and first snow-free patches had appeared. Snow depth varied between 0 cm and 30 cm at the Sodankylä site and between 0 cm and 60 cm at the Saariselkä site. Additionally, more snow-free pixels were found in Sodankylä than in Saariselkä. The measurement set-up followed the earlier campaign. The measurements were carried out under direct illumination (cloud cover 0/8) in Sodankylä and under diffuse illumination (cloud cover 7/8) in Saariselkä. The reflectance level was obtained by applying a real-time fibre optic downwelling irradiance sensor (FODIS) (Heinilä et al. 2019b). In both campaigns the Oxford Technical Solutions RT4000 GPS/INS was utilized to provide high accuracy position measurements with low drift rates.

The imaging spectrometer data was radiometrically and geometrically corrected by using the SPECIM's CaliGeo tool in the

ENVI software. Measurements from Saariselkä were additionally corrected with the digital elevation model KM10 (Finnish national digital elevation model by the National Land Survey) with a pixel size of $10 \times 10$ m$^2$ and elevation resolution of 1.4 m. The bands 555 nm, 645 nm, 858.5 nm and 1640 nm were extracted from the original spectra by using the band specific FWHM (full width at half maximum) criterion corresponding to MODIS bands. For these bands the original 80 cm resolution data was filtered with mean filter using 12x12 window corresponding to pixel size of $10 \times 10$ m$^2$.


As an example, Fig. 12 shows reflectance values on 5 May 2011 observed over different land cover types during partial snow cover along the AisaDUAL flight line. At the very end of the spring melting the observed reflectances are relatively low in all land cover types even with 50-60% snow patchiness.

(a)                                                                      (b)

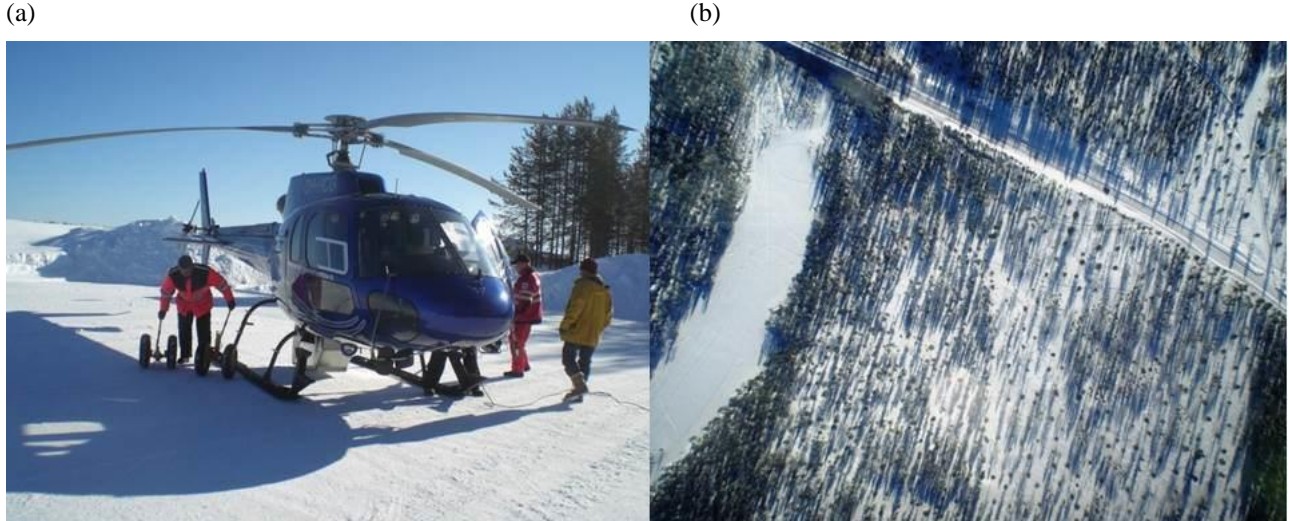

**Figure 11. a) In March 2010 the AISA acquisition was made from a helicopter. The AISA sensor was attached in a box mounted on the bottom of the helicopter. In the bottom of the box was a hole for the sensor. b) The photo taken from the helicopter on 21 March 2010.**

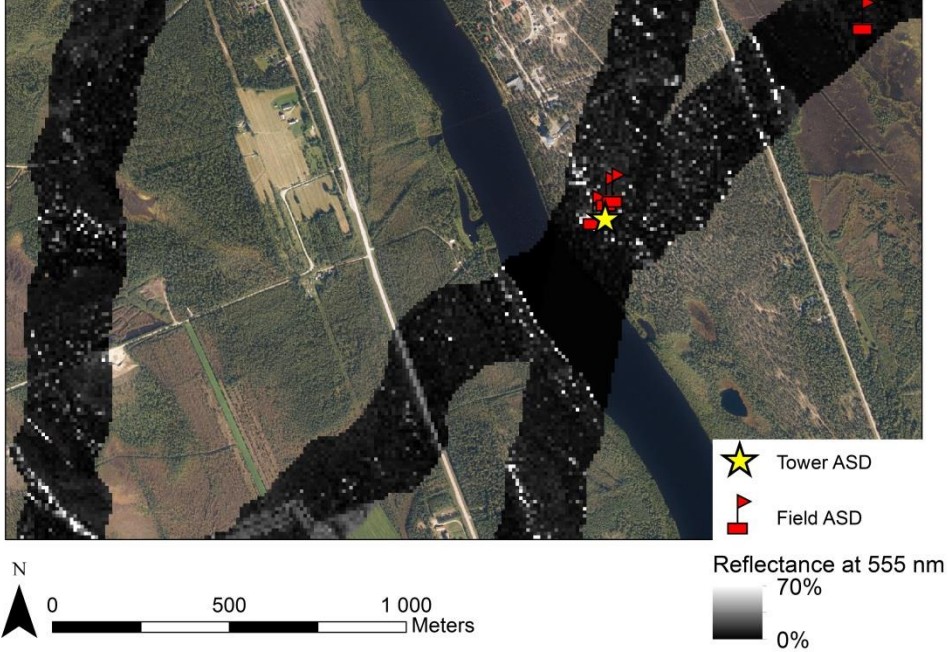


**Figure 12. Airborne spectrometer reflectance at band 555 nm on 5 May 2011 at 10 m resolution. In the background an ortophoto from summer time conditions © National Land Survey of Finland, 12/2018.**

### 3.5. Reference measurements

With each case of measured spectra of snow and open ground targets described above, reference in situ measurements and observations of weather conditions, location characteristics and snow properties were conducted, to help to interpret the changes seen in the measured spectra (Table 2). The portable field measurements and the accompanied in situ data serve as reference information also for the airborne measurements (overlapping in time). The prevailing weather conditions were logged while making the reference observations. These included measurement of air temperature at two meter height and observation

of cloud cover in octas. For the mast measurements, cloud cover (in octas, Vaisala CT25K laser ceilometer), air temperature at 2 m (10 min average) and wind speed in gust at 22 m (10 min maximum) from an automatic weather station were accompanied with the measured spectra. Since the measurement automatization in 2012, the Δ-value, describing the illumination change between the target and the reference measurement, were also added as reference data for the measured spectra. The mast measurement area was photographed with wide-angle digital camera after each spectral measurement for

target characteristics description. Although no other specific reference data were acquired for the mast observations, the extensive collection of automatic and manual in situ observations from the IOA site are available, including weekly snow pit measurements collected along the Sodankylä Manual Snow Survey Programme (Leppänen et al., 2016). As the field measurements were also conducted in different terrain, some information was collected about the surroundings, namely land cover type and snow patchiness. The snow properties logged in the reference data were: snow temperature from different

depths, layering of snow, geometric snow grain size, type and specific surface area (SSA), observed and measured snow water content, snow density and impurities/organic litter in snow. The methods and parameters varied somewhat between the laboratory and field reference data. The reference data for laboratory and field measurements are summarized in Table 3.

**Table 3. Supplementary parameters measured from each snow type condition represented in the portable field measurements and in the snow laboratory experiments.**

| Supplementary data | Unit | Laboratory reference<br>Method and accuracy | Field reference<br>Method and accuracy |
|---|---|---|---|
| Coordinates | [°] | | Latitude and longitude (WGS 84)/ ± 10 m |
| Date | | | Gregorian date |
| Hours | | | Time GMT +2 |
| Minutes | | | Time GMT +2 |
| Land cover | | | Nine classes: 1) open(heath); 2) open mire; 3) pine forest; 4) spruce forest; 5) mixed forest (coniferous dominant); 6) sparse forest; 7) fell; 8) snow on lake ice; 9) grassland/agricultural field |
| Air temperature | [°C] | Measured at free air in shadow/ ± 0.2°<br>TH310 thermometer (Mil-waukee Electronics Kft., Szeged, Hungary) | Measured at free air in shadow/ ± 0.2°<br>digital thermometer (unk.) |
| Cloud cover | | Estimated visually in octas.<br>[0/8 … 8/8]<br>0/8 clear sky, 8/8 full cloud cover | Estimated visually in octas.<br>[0/8 … 8/8]<br>0/8 clear sky, 8/8 full cloud cover |
| Snow depth | [cm] | 3 measurements at least 1 m apart/ ± 1cm<br>(wooden snow measurement stake) | 3 measurements at least 1 m apart/ ± 1cm<br>(wooden snow measurement stake) |
| Snow patchiness | [%] | NA | Estimated visually in surrounding area/ ± 10% |
| Snow temperature | [°C] | Surface and then every 10 cm/ ± 0.2°<br>TH310 thermometer (Mil-waukee Electronics Kft., Szeged, Hungary) | Measured from two depths, 5 cm and at half of the depth/ ± 0.2°<br>digital thermometer (unk.) |
| Soil surface temperature | [°C] | NA | Measured at snow-soil interface/ ± 0.2°<br>digital thermometer (unk.) |
| Layering | [cm] | Based on hardness, grain size and density differences/ ± 1cm | NA |
| Geometric snow grain size | [mm] | Typical maximum grain diameter (estimated from macrophotographs taken against a 1 mm grid)/ ± 0.25 mm | Visual estimate against a millimetre grid/ ± 0.5 mm |
| Snow grain type | | According to Fierz et al. (2009) | Six classes: 1) fine separated crystals; 2) metamorphosed separated crystals; 3) clustered crystals; 4) almost slush; 5) slush; 6) ice layer |
| Snow water content | | Five classes: 1) dry; 2) moist; 3) wet; 4) very wet; 5) slush according to Fierz et al. (2009) | Five classes: 1) dry; 2) moist; 3) wet; 4) very wet; 5) slush according to Colbeck et al. 1990, now Fierz et al. (2009) |
| Snow density | [g/cm3] | Sampled every 5 cm by snow density cutter | Sampled every 5 cm by snow density cutter |
| SSA (IceCube, A2 Photonic Sensors, Grenoble, France)* | [m2/kg] | Sampled every 3 cm | NA |
| Snow wetness (Snow Fork, Toikka Ltd. | [%] | Sampled every 10 cm | NA |

| | | | |
|---|---|---|---|
| Engineering, Espoo, Finland) ** | | | |
| Snow density (Snow Fork, Toikka Ltd., Engineering, Espoo, Finland) ** | [g/cm3] | Sampled every 10 cm | NA |
| Forest litter/impurities in snow surface | | Two classes: 1) no litter; 2) litter visually estimated from snow surface | Two classes: 1) no litter; 2) litter visually estimated from snow surface |

* Gallet et al. (2009)

    ** Sihvola and Tiuri (1986)

## 4. Discussion

### 4.1. Error and uncertainty

The target scene reflectance inside the satellite footprint, recorded by a remote sensing instrument, is a combination of spectral information of several endmembers which complicates the data interpretation. Thus knowledge of the spectral reflectance characteristics of the target endmember (e.g. snow) as well as the combined effect of several contributing endmembers (e.g. forest and open ground) is needed. Data of the same quantity at several scales allows accumulation of understanding from reflective properties of an individual tree branch or snow type to scene reflective properties observed at mast-scale to a scale of an optical satellite footprint of several hundred meters. As the sources of error and uncertainty are variable, data at multiple scales also benefits the recognition and quantification of inaccuracies in the remotely sensed information.

The spectroscopy measurements are affected by manifold factors yielding to error and uncertainty in the observations and therefore complicating the understanding of the effects of the measured target on the propagation of electromagnetic radiation. These factors stem partly from the instrument characteristics and partly from prevailing conditions. Spectral and radiometric calibration and stability characterization are required to address the effects of the instrumental uncertainties. Optimal sampling procedure appropriate for the considered application should be chosen and the common measurement protocols and standards followed. The imperfections in the reflectance calibration need to be recognized and the effect of uncontrolled factors, such as changing illumination conditions, should be minimized and documented (Hueni et al., 2017). The measurements of reflectance properties of snow and snow-free ground targets in different spatial scales have enabled the estimation of the systematic error involved in satellite algorithms for snow retrieval (Salminen et al., 2018). In order to use the subordinate scale, the relevant error sources need to be identified and preferably quantitatively estimated. Here the sources of measurement error and uncertainty of the described datasets are discussed.

In the laboratory conditions the measurements are highly controllable. The external error sources can therefore be minimized. In such conditions the precision of the measurements can be estimated based on the repeated measurements of a reference Spectralon panel. The integrated precision is determined by (Hannula and Pulliainen, 2019):

$$S(\lambda, \theta) = \frac{1}{N} \sqrt{L_{s1}(\lambda)^2 + L_{s2}(\lambda)^2 + \cdots + L_{sN}(\lambda)^2} \qquad (3)$$

where $S$ is the precision of the calibration with light zenith angle $\theta$, and $L_{sN}$ the standard deviation of the Spectralon radiance acquisitions (n=10–30) at wavelength $\lambda$ for individual reference panel measurements 1 - N. The precision of the laboratory measurements was lowest in the detector edges (1000 nm and 1800 nm) and at the both ends of the spectral range. Excluding these areas, the precision varied roughly between $2.0*10^{-6}$–$8.0*10^{-6}$ W m$^{-2}$ sr$^{-1}$ nm$^{-1}$ for 470–830 nm and 1200–1790 nm and

between $8.0*10^{-6}–2.0*10^{-5}$ W m$^{-2}$ sr$^{-1}$ nm$^{-1}$ for the last detector (1800–2300 nm) due to lower signal-to-noise ratio (SNR). The precision values describe the integrated inaccuracy due to e.g. instrument instability (in all measurements the instrument was left to warm-up 30 minutes at minimum) and lamp irradiance variability (+/- 0.1 %). In addition, there may be other known and unknown systematic and random sources of error such as possible stray light from any reflecting surface and inaccuracies in the desired measurement geometry. During the portable field observations, real-time reflectance spectra were collected (no separate Spectralon radiances were saved), whereas the mast-borne and airborne observations represent only one measurement acquisition. Thus, similar estimates of the measurement precision for field, mast, or airborne conditions were not possible to define. Earlier studies have shown that the uncertainty characterizations made in laboratory conditions can remarkably differ from those derived in the field (Anderson et al., 2011).

Correct calibration is essential to obtain high quality reflectance data. As such, the uncertainty at all scales of the data record presented here is related to the uncertainty in the calibration. In laboratory, this is mostly related to the imperfect Lambertian characteristics of the Spectralon panel. Sandmeier et al. (1998) and Rollin et al. (2000) have shown that Spectralon panels have anisotropic reflectance characteristics depending on view and illumination geometry. This causes some systematic (+/-) uncertainty in the absolute reflectance values in this data record. In the airborne, mast-borne and portable field data the uncertainty and error in the calibration is added by the possibility of the panel degradation. The degradation level of the reference panel can be estimated by calibration tests against a laboratory standard repeated in time (see 2.2). For successful calibration, it is also important that the panel is absolutely horizontally aligned. In portable field observations, a tripod with a bubble level was used to reduce the error from panel alignment which, according to earlier demonstrations (Hueni et al., 2017) should lead to a deviation of less than 1° from the horizontal alignment.

The measurements in field conditions, including the mast-platform and airborne measurements, are more susceptible to changes in the external conditions. The field measurements are affected by the naturally varying illumination, atmospheric composition, and measurement geometry, but also by the possible reflective or obstructive objects in the measurement surroundings. In field measurements the observed target (directional) radiation may change without any changes in the target properties if the distribution of irradiation over the hemisphere is changed (Kriebel, 1976). This is due to the anisotropic reflective properties of natural surfaces. Under clear sky conditions, changes in the incident irradiance are governed by the changes in sun zenith angle and the optical depth of the atmosphere (Goetz, 2012; Kriebel, 1976). To minimize these effects the frequency of Spectralon measurements should be adjusted according to the stability of the illumination (see 2.2) and measured near or at the same location as the target (Goetz, 2012; Mac Arthur and Robinson, 2015). In ideal case the measurements are executed around the local noon if the purpose is not to study the effect of changing illumination conditions, as e.g. in the mast-borne measurements. Accordingly, any nearby objects, including the observer, will affect the spectral measurements by blocking part of the diffuse irradiance and on the other hand by reflecting the down-welling (direct and diffuse irradiance) and up-welling (reflected from ground) radiance towards the target (Kimes et al., 1983). If the location of

these objects remain the same in relation to the target and the Spectralon no error is produced, but this is rarely the case in the field. This speaks in favor of fixed installations, such as mast-platform where at least the measurement setup itself remains unchanged (Hueni et al., 2017). In the portable field measurements, the tripod with the extended arm obscured a part of the diffuse skylight illuminating the target. This effect has not been quantified nor corrected in our measurements. The airborne measurements are affected by the adjacency effect in the heterogeneous areas where top of atmosphere (TOA) radiance is decreased over bright pixels and increased over dark pixels (Otterman and Fraser, 1979). This can be reduced by calibrating the TOA radiances using surface radiances from the same target as was done for the AISA radiances in March 2010. The effect of the external factors may become mixed with the reflectance variability caused by the target properties, such as snow characteristics, and thus make conduction of field measurements complex. In field, the uncorrected irradiance levels and other external sources of error together with the BRDF (bidirectional reflectance distribution function) characteristics of the target may compensate each other, resulting in less variable reflectance (Sandmeier et al., 1998). These interactions are target specific and are typically hard to predict (Sandmeier et al. 1998). Thus, the reflectance observed in laboratory conditions can be more reliably interpreted to be originating from the target's properties.

The measurement scale needs to be taken into account when interpreting the results as the chosen sensor to target distance combined with spatial heterogeneity of the target may yield into very different outcomes (Milton et al., 2009). This was demonstrated in Fig. 6 where a shift of the pine and spruce sample boxes under the measurement head was followed by a clear change in the target reflectance. Accordingly, change in sun azimuth angle over an asymmetric surface (such as forest canopy) without change in the target properties will yield into different reflectance value (Kriebel, 1976). Thus, the representativeness of the dataset has to be judged in respect of the temporal and spatial sampling and the aim of the study. Instrument characteristics may introduce uncertainty. Photodiode detectors utilized in spectroradiometers have temperature dependent sensitivities (Hueni and Bialek, 2017; Starks et al., 1995). In the mast-borne and in the portable field measurements the spectroradiometer was placed inside an insulated box for protection and to decrease the variability of the ambient temperature. The spectroradiometer utilized in laboratory and portable field measurements has been regularly calibrated at the manufacturer. The mast-borne spectroradiometer has been calibrated on a less regularly basis, but the changes of the instrument responsivity have been monitored by yearly laboratory tests to reveal any changes in the instrument behaviour. Some instrument characteristics are hard to determine in detail. ASD spectroradiometer FOV have shown to differ from the nominal FOV reported by the manufacturer and the sensor responsivity to be nonuniform within the FOV (Mac Arthur et al, 2012). This complicates the understanding of the relationship between the observation and the target in heterogeneous areas (Hueni et al., 2017). These examples illustrate the complexity of the factors affecting the (field) spectroscopy measurements and highlight the need for comprehensive metadata of the measurement sites to assist the data interpretation.

## 4.2. Reflectance of same targets measured by different platforms

Comparison of observations collected by different platforms is not always straightforward. Fig. 13 presents surface reflectance
spectra observed at different scales for snow covered lake ice (a) and forest measurement area of the mast-borne platform during dry snow conditions (b). For Fig. 13b. the mean of pine branch reflectance measurements, measured in a laboratory, is also shown. The motivation behind the comparison of measurements collected at different scales is to understand how the band reflectance value measured for a coarse resolution remote sensing image pixel is composed for different types of heterogenous landscapes. The same motivation behind these studies may rise problems in future data analysis. For a homogeneous area with
direct and stable illumination conditions comparing measurements observed from a height of 800 m and 45 cm and with a spatial resolution of 10 m and 20 cm may give information, for example, from the success of the atmospheric correction. In Fig. 13a there are some differences in the snow reflectance observed at different scales, but the airborne values still fit within the standard deviation observed at ground in the portable field measurements. When more heterogeneous surfaces, such as forested area in Fig. 13b are compared, even small differences in the view angles (nadir for airborne AISA and 11° for mast-
borne observations) yield in to differences, which although giving information about the effects of these differences, also complicate the comparison as it is more difficult to distinguish the effect of one case from another. From the different view angles in Fig. 13b it follows that the forest cover for the same area observed by the airborne platform is 40 % whereas it is 48 % when observed by the mast-borne platform. The proportional areas of shadowed surfaces and whether the sensor is mostly seeing pine branches or both tree branches and tree trunks are also affected by the view angle. Point-wise field measurements
and temporally restricted laboratory observations can be, however, successfully used to characterize the behavior of larger scale measurements via modelling (Niemi et al., 2012). Many studies have researched the spatial representativeness of observations collected at different scales for both homogeneous and heterogeneous surface types (e.g. Román et al., 2009; Wang et al., 2014).

(a)

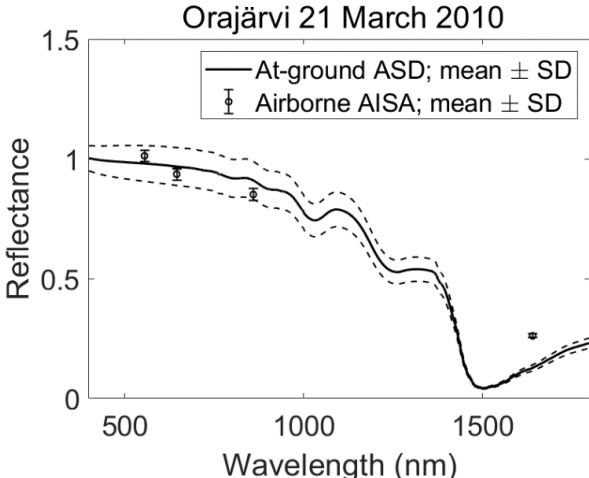

(b)

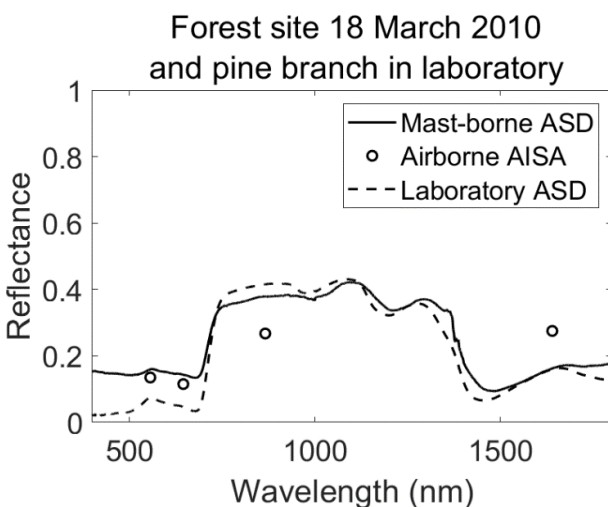

**Figure 13. a) Snow reflectance on lake Orajärvi measured by airborne (AISA) and portable field platforms. b) Reflectance of the forest measurement area observed by the mast-borne (spectra) and airborne (four bands) platforms during dry snow conditions and mean of pine brach measurements observed in a laboratory.**

## 4.2. Examples of data usage

In the laboratory experiments the aim was to characterize the variation of the spectral reflectivity of pine and spruce and different snow types (i.e. spectral endmembers) with controlled illumination, a characteristic which cannot be reached in field conditions. Since the pine and spruce sample reflectances at this scale can significantly change depending on the orientation of the target in relation to the measurement head, a number of observations with varying orientations were taken to describe the average variance. Both laboratory and field observations can describe only part of the spatial and temporal variability in the targets spectral reflectance as only specific number of measurements at some specific times can be measured. With continuous mast measurements a time series of reflectance spectra of the same target area can be constructed offering data to study the changes in the spectra of a specific land cover type in varying illumination and atmospheric conditions and with seasonally varying target characteristics. In comparison, the airborne data provides the variability between several boreal land cover types. Scaling upward with mast- and airborne data records gives one more link between the remotely sensed and point-wise field observations. The presented data record can be considered representative as it is measured with various temporal and spatial resolutions and has the specific advantage of being coincident in time and from the same locale.

The data record has been utilized in several feasibility studies of satellite snow covered area mapping most of them focusing on forested areas. The changes in portable field spectra due to snow properties were studied by Salminen et al. (2009) and Niemi et al. (2012). They showed that snow wetness had strong effect on the forward scattering due to increase of the effective grain size in the optical region (Wiscombe and Warren 1980). This produced high variability in the reflectance spectrum (Niemi et al., 2012). Wet snow transmits light more efficiently and therefore, during the spring melting conditions, snow depth starts to play a more significant role in altering the reflectance. The mean snow reflectance can drop from 1.00 to 0.7, when a threshold of 20 cm snow depth is crossed (Salminen et al., 2009).

Salminen et al. (2009) used their own point-wise portable field spectroradiometer measurements to statistically characterize the variability of boreal ground reflectance and mast-borne time series to study the comparability of point-wise and scene reflectance measurements aiming at optimal band selection and assessment of accuracy when applying the SCAmod method, an algorithm for FSC detection. They concluded that ground reflectance variability can induce errors up to 10–12 % in SCA estimations and suggested the use of wavelengths 400–480 nm for SCAmod (and other similar) methods for the best detection of snow. The work was continued by Niemi et al. (2012) who utilized the mast-borne observations of forest and forest opening to investigate the boreal forest scene reflectance behaviour by means of NDSI (normalised difference snow index), NDVI (normalized difference vegetation index) and MODIS bands during springs of 2010 and 2011. In forest opening the band indices were well functioning but at the forest scene were strongly affected by the illumination geometry. The study of the spectral index behaviour was continued by investigating the linkage between the scene reflectance and the forest canopy characteristics (coverage, tree height) by concurrent use of field, mast-borne and airborne spectral measurements and LIDAR

data (Heinilä et al., 2014, 2019b). Airborne reflectances from snow-covered surfaces were shown to be highly dependent on forest characteristics. In Pulliainen et al. (2014) the mast-borne measurements from 2010 and 2013 and airborne data record from 2010 were once again utilized to test a zeroth order radiative transfer approach for snow monitoring from optical remote sensing data. By means of these data records, the spatial and temporal variability of boreal forest reflectance could be investigated and the model validated at several different scales.

In forthcoming research the mast-borne data record will be further utilized to analyse the representativeness of the mast measurements for the larger boreal forest area in FMI-ARC surroundings and to assess the feasibility of the latest optical satellite data provided in higher, 10–30 meter spatial resolution. Spectral data at multiple scales offer a possibility to assess the effect of atmospheric correction applied in remote sensing data processing. Meteorological observations as well as manually and automatically measured snow properties from FMI-ARC have also been used to drive and evaluate snow models (Essery et al., 2016; Menard et al., 2019). Driving models benefits from (hemispherical) albedo measurements but also directional-hemispherical reflectance observations may be an interest for the snow modelling community. Although the collection and analysis of spectral data record has been driven by the aim to improve optical snow mapping methods, multiple other possibilities for data usage exist. The mast-borne data can serve as a direct validation or cross-reference information for Unmanned Aerial Vehicle (UAV) -borne spectral measurements and the spectral range (Table 1) is valid for phenology or vegetation spring green-up studies.

## 5. Data availability

The data record is made available through a community in Zenodo repository service (https://www.zenodo.org/communities/boreal_reflectances) (Hannula et al., 2019). Each dataset of a distinct scale has its own unique Digital Object Identifier (DOI): laboratory: 10.5281/zenodo.3580078 (Hannula and Heinilä, 2018a), field: 10.5281/zenodo.3580825 (Heinilä et al., 2019a) , see also https://ckan.ymparisto.fi/dataset/spectrometer-measurements-of-snow-and-bare-ground-targets-and-simultaneous-measurements-of-snow, mast-borne: 10.5281/zenodo.3580096 (Hannula and Heinilä, 2018b), and airborne: 10.5281/zenodo.3580451 (Heinilä, 2019a) and 10.5281/zenodo.3580419 (Heinilä, 2019b). Two DOI numbers for AISA datasets were created due to differing data processing methods in 2010 and 2011. The ASD spectra as well as the accompanied reference measurements are organized in ASCII-files the metadata information attached in the file header (laboratory, mast-borne and airborne) or as a separate metadata file (portable field). The airborne AISA reflectances are provided as geolocated GeoTIFF files. Digital images for the mast-borne measurement scenes are organized in yearly folders and packed into a zip file. For some spectra the digital images are missing due to technical problems. In 2010–2011, when the mast-borne measurements were still collected manually, the time gap between the measurement and the digital image varies. However, images temporally far apart are sometimes included in the dataset as they still give some information about the measurement target for the user. The image file names have an associated time tag and an indication of the

measurement target in format YYYY-MM-DD_HHMMSS_forest/open. For further information, contact details are provided along with each dataset. The laboratory, portable field and airborne datasets have undergone quality check to include only the good quality spectra or band reflectances. However, the users are encouraged to consider the sources of error and uncertainty discussed in chapter 4.1. Only very robust quality check, leaving out the most erroneous measurements, has been conducted on the mast-borne dataset. Any further filtering is left for the user as best seen for the application. Users are also encouraged to give feedback on any issues with the datasets.

## 6. Conclusions

In order to establish new and improved optical snow mapping methods for boreal forested areas, detailed surveys of satellite scene reflectance contributors are required, as the relatively large satellite footprint may contain both fractional snow and forest cover. The spectral reflectance data record described here contains spectral observations of the main components (i.e. spectral endmembers) of a boreal landscape during spring: snow (dry, wet, shadowed), forest ground (moss, lichen) and forest canopy (spruce and pine, branches) corresponding to atmospherically corrected estimate of surface reflectance from satellite data. The data record contains comparable observations at laboratory, field, mast-borne and airborne scales the last three scales overlapping in time. In addition, the collection includes reference data collected in situ, along with spectral observations.

The main experimental site for data collection in Sodankylä, northern Finland, and the collection and measurement systems for each scale of data record were described in detail and data examples were given. The possible sources of error and uncertainty were discussed and estimated. So far, the data record has been used for various scientific studies most of them focusing on the improvement of snow cover detection in forested areas. However, data record at various scales offer numerous other possibilities for data usage such as cross-reference information for UAV-borne spectral measurements or phenology and vegetation spring green-up studies.

### Author contribution

HRH was responsible for the planning, coordination and conduction of the snow laboratory measurements, post-processed the mast-borne data for the period 2012–2018 and wrote most of the chapters 3.1, 3.3, 4–7, and was responsible for the overall writing work of the study as well as the submission process. KH was responsible for the coordination and execution of the pine/spruce branch laboratory experiments, post-processed the mast-borne data for the period 2010–2011 and took part into the collection of portable field spectral measurements. She post-processed the airborne data and wrote most of the chapter 3.4. KB took part into the collection of the portable field spectral measurements and post-processed the data, wrote most of the chapter 3.2 and generated the original idea of the article. OPM took part in the collection of the portable field spectral

measurements and the writing work of the study. MS wrote most of the chapters 1 and 2. JP was behind the measurement idea of most of the datasets described and acted as scientific supervisor of the manuscript.


## Competing interests

The authors declare that they have no conflicts of interest.

## Disclaimer

Data record has undergone some preliminary quality check but any further quality control is left for the user as best seen for the purpose.

## Acknowledgements

This work has been supported by Maj and Tor Nessling Foundation (201500276, 201600013, 201700417), Väisälä foundation,
The European Commission Life+ program funded SnowCarbo project (ENV/FIN/000133), Envibase (Ministry of Finance, Finland), the Academy of Finland SA-CarbArc (285630), SPECIM (Spectral Imaging Ltd.) and the Airborne Imaging Spectroscopy Application and Research on Earth Sciences (AISARES) graduate school of the University of Helsinki.

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
