# Peer review of "Laboratory, field, mast-borne and airborne spectral reflectance measurements of boreal landscape during spring"

_Earth System Science Data, 2019_

## Referee Comment (RC1) · Anonymous Referee #1 · 11 Sep 2019

The paper describes the data collection of spectral reflectances conducted mainly in the Sodankylä region over multiple platforms. The measurements aim to be representative of different snow and weather conditions (e.g. wet or dry snow for the former, clear or covered for the latter) and were organised such that measurements from different platforms were overlapping.

The manuscript is well written and easy to read from start to finish. The laboratory measurements, field campaigns and instrumentation as well as potential sources of errors and uncertainties are thoroughly described. The multi-platform measurements seem to have been very well coordinated and organised. As a consequence I would

change very little of what is currently in the manuscript and only have minor comments/clarifications.

My two main comments therefore do not concern what is in the manuscript, but what is missing from it. Firstly, one of the most interesting aspect of this study is the availability of data looking at the spectral reflectance of the same surfaces but with different instruments. The manuscript incomplete and will remain so unless a section (1) compares the reflectances obtained on different platforms on overlapping dates over the same surfaces (2) discusses the implications of the differences, bearing in mind future users (3) plots from multiple platforms showing spectral reflectances on overlapping days over the same surfaces are added.

Secondly, the data in zenodo are well organised, but their (justified) discretization into platform and scale means there is a large number of files for potential future users to wade through. It would be useful, for each platform and scale, to include representative plots of each dataset in zenodo to have a quick visualisation of the sort of data available. It is one thing to make data available, it is another to make them user-friendly and, as such, re-usable. The manuscript describes invaluable datasets that should be published and used, and I trust that adding such quick visualisation of the data through these plots will help make these datasets more user-friendly.

I trust that the manuscript will be fit to publish when the above suggestions and minor comments below are addressed.

Minor comments: Line 134 - Sodankylä is most probably taiga snow. The Sturm snow cover classification system has been accepted as the standard in our field for a long time, but it is perhaps time we acknowledge its limitations: the European Alps are, after all, classified exclusively as maritime. While it is not the task of this manuscript, I am confident the authors are very familiar with the type of snow in Sodankylä and could therefore rely on their own expertise, rather than on a classification relying exclusively on measurements from Alaska, to describe it.

L149 - Could there be a quick explanation of what Spectralon is?

L395 - Minor difference, but I think changing the start of the sentence to "As an example, Figure 11 shows reflectance values on 5 May 2011 observed over et." would make it sound less like Fig 11 is a random example not even used in the campaign. Adding the exact date to Figure 11, rather than just "May 2011", would also help clarify.

Table 2: This is a big table and it is easy to lose some information. If possible, a Gantt chart or something similar showing the multiple platforms and overlapping dates would make it easier to see which measurements from which platforms are overlapping.

Figure 9: Is the label on the y axis correct? These are not MODIS band 4 reflectance measurements, but mast-spectroradiometer measurements to match MODIS band 4. This should be clearer. Conclusion: Data from Sodankylä are also being used for driving and evaluating snow models (Essery et al., 2016, gi-5-219-2016) and Earth System Models, notably as part of ESM-SnowMIP (Menard et al., 2019, essd-11-865-2019). It may be worth mentioning that adding albedo measurements to these datasets would be invaluable to the snow modelling community.

---

## Author Comment (AC1) · 12 Nov 2019

GENERAL COMMENTS

1) General comments from the referee

The paper describes the data collection of spectral reflectances conducted mainly in the Sodankylä region over multiple platforms. The measurements aim to be representative of different snow and weather conditions (e.g. wet or dry snow for the former, clear or covered for the latter) and were organised such that measurements from different platforms were overlapping.

[Figure]

The manuscript is well written and easy to read from start to finish. The laboratory measurements, field campaigns and instrumentation as well as potential sources of errors and uncertainties are thoroughly described. The multi-platform measurements seem to have been very well coordinated and organised. As a consequence I would change very little of what is currently in the manuscript and only have minor comments/clarifications.

My two main comments therefore do not concern what is in the manuscript, but what is missing from it. Firstly, one of the most interesting aspect of this study is the availability of data looking at the spectral reflectance of the same surfaces but with different instruments. The manuscript incomplete and will remain so unless a section (1) compares the reflectances obtained on different platforms on overlapping dates over the same surfaces (2) discusses the implications of the differences, bearing in mind future users (3) plots from multiple platforms showing spectral reflectances on overlapping days over the same surfaces are added.

Secondly, the data in zenodo are well organised, but their (justified) discretization into platform and scale means there is a large number of files for potential future users to wade through. It would be useful, for each platform and scale, to include representative plots of each dataset in zenodo to have a quick visualisation of the sort of data available. It is one thing to make data available, it is another to make them user-friendly and, as such, re-usable. The manuscript describes invaluable datasets that should be published and used, and I trust that adding such quick visualisation of the data through these plots will help make these datasets more user-friendly.

I trust that the manuscript will be fit to publish when the above suggestions and minor comments below are addressed.

2) Author's response on general comments by the referee

We thank the referee #1 for a thorough reading of the manuscript and bringing out the current weaknesses. We agree that without comparison of reflectance from different

platforms (from overlapping times and same surface targets) the possibly most valuable part of the data record remains underemphasized. Accordingly, we think that adding some plots of data on Zenodo would benefit the user by giving an instant idea of the dataset content.

3) Author's changes in manuscript

According to the suggestions by the referee, we will add a figure representing reflectance of the same target(s) but measured on different scales. Utilizing the figure we will compare the measurements from different platforms and discuss the implications of the scale.

———————————————————————————————

MINOR COMMENTS

1) Minor comment from the referee

Minor comments: Line 134 - Sodankylä is most probably taiga snow. The Sturm snow cover classification system has been accepted as the standard in our field for a long time, but it is perhaps time we acknowledge its limitations: the European Alps are, after all, classified exclusively as maritime. While it is not the task of this manuscript, I am confident the authors are very familiar with the type of snow in Sodankylä and could therefore rely on their own expertise, rather than on a classification relying exclusively on measurements from Alaska, to describe it.

2) Author's response and 3) changes in manuscript

We will rely on our own knowledge rather than general classification when we will revise the manuscript.

———————————————————————————————

1) Minor comment from the referee

L149 - Could there be a quick explanation of what Spectralon is?

2) Author's response and 3) changes in manuscript

Yes, there could be a short explanation, and this would benefit especially users outside the spectroscopy community. The explanation will be added.

—————————————————————————————————

1) Minor comment from the referee

L395 - Minor difference, but I think changing the start of the sentence to "As an example, Figure 11 shows reflectance values on 5 May 2011 observed over et." would make it sound less like Fig 11 is a random example not even used in the campaign. Adding the exact date to Figure 11, rather than just "May 2011", would also help clarify.

2) Author's response and 3) changes in manuscript

We will reformulate the sentence as suggested by the referee.

—————————————————————————————————

1) Minor comment from the referee

Table 2: This is a big table and it is easy to lose some information. If possible, a Gantt chart or something similar showing the multiple platforms and overlapping dates would make it easier to see which measurements from which platforms are overlapping.

2) Author's response and 3) changes in manuscript

We admit the Table 2 is large. However, we would like to keep it to have all the information from different platforms in the same place. In addition, we will compile a Gantt chart (or similar) as suggested to easily see the overlap and time ranges of the different measurements.

—————————————————————————————————

1) Minor comment from the referee

Figure 9: Is the label on the y axis correct? These are not MODIS band 4 reflectance measurements, but mast-spectroradiometer measurements to match MODIS band 4. This should be clearer.

2) Author's response and 3) changes in manuscript

The y-axis label truly is misleading. We will change the label.
* * *
1) Minor comment from the referee

Conclusion: Data from Sodankylä are also being used for driving and evaluating snow models (Essery et al., 2016, gi-5-219-2016) and Earth System Models, notably as part of ESM-SnowMIP (Menard et al., 2019, essd-11-865-2019). It may be worth mentioning that adding albedo measurements to these datasets would be invaluable to the snow modelling community.

2) Author's response and 3) changes in manuscript

This is a good point and we will add it as one opportunity for the data usage in the conclusion.

---

## Referee Comment (RC2) · Anonymous Referee #2 · 13 Nov 2019

Remarks to the Authors

Review of "Laboratory, field, mast-borne and airborne spectral reflectance measurements of boreal landscape during spring" by Henna-Reetta Hannula et al.

Earth System Science Data., Manuscript Number: essd-2019-88

————————————————————————————————————————

General comments:

In this paper, the authors introduce and describe detailed spectral reflectance data for some types of snow, forest canopy, snow-on-canopy, snow-free patches, etc obtained

from laboratory, field, mast-borne, and airborne optical measurement systems. The study areas are mainly located in the Arctic region of Finland. The main purpose of the data acquisition is to provide basic information for the development of new and improved optical snow mapping methods for boreal forested area using satellite data. This kind of remote sensing study is very important recently, because seasonal snow physical conditions, which would affect water resources around the area for example, are rapidly changing due to the ongoing global warming. This reviewer found the data acquisition methods and procedures described in this paper are solid, and presented data are reliable in my opinion. Overall, this paper is detailed, well written, and structured; however, this reviewer suggests the following points to be considered before the publication.

Please note that page and line numbers are denoted by "P" and "L", respectively.

––––––––––––––––––––––––––––––––––––––––––––––––––––––––––––––––––––––

Specific comments (major)

P. 16, L. 262: How about showing the standard deviations mentioned here in this paper? I think this information, which show accuracy of the data, is very important.

P. 16, L. 276: What do the authors think about the effects of the tripod on the measured reflectance data? Please discuss briefly here.

P. 20, L. 346 ∼ 347: Please explain more in detail about the resampling procedure (about the choice of a weighting function, etc).

––––––––––––––––––––––––––––––––––––––––––––––––––––––––––––––––––––––

Specific comments (minor)

P. 1, L. 16: For the location of Sodankylä Arctic Space Centre, please indicate altitude of the site together with the Lat Lon information.

P. 2, L. 33: Maybe, citing the latest version of the SWIPA report (AMAP, 2017) instead

of the previous report (AMAP, 2011) would be better here.

P. 2, L. 46: It is better to explain the definition of "spectral endmembers" here especially for non-specialist readers.

P. 2, L. 50 ∼ 51: Please consider citing the papers by Aoki et al. (2000), Carmagnola et al, (2014), and Tanikawa et al. (2014).

P. 3, L. 72: What do the authors mean by "samples"?

P. 3, L. 95 ∼ 97: Please indicate spectral resolutions of these data here.

P. 4, L. 111: What is fjell?

P. 5, L. 131: What do the authors mean by "upper atmosphere's perspective"?

P. 6, L. 134: For snow classification, it is better to refer the international snow classification (Fierz et al., 2009).

P. 6, L. 148: Please detail more about the white reference standard used in this study (e.g., manufacturer, location of the manufacturer, and type).

P. 9, L. 211 ∼ 213: Maybe, referring to photos in Figures 4 and 6 here would be very helpful for readers.

P. 10, Table. 1: I think there is a higher-resolution version of ASD Field Spec Pro. Do the authors think using a standard version of Field Spec Pro is enough for the purpose of this study (the purpose of the data is summarized well in the second paragraph of the Introduction section)?

P. 10, Table. 1: This reviewer is interested in how the authors determined the distances from the sensors to targets especially for the Lab and Portable cases. Please explain.

P. 11, Table 2 (The following comment is related to the previous comment for "P. 3, L. 72"): If I understand the meaning of "samples" correctly, I would like to know why the numbers of samples per target in Table 2 vary from date to date. I would make the

number the same throughout the study period when I do this kind of measurements.

P. 19, L. 317: Are "hours 10, 12, and 14" in local time?

P. 22, Figure 10a: Please mention where the sensor is attached in the helicopter.

P. 24, L. 419 and Table 3: Please consider adding "geometric" before "snow grain size" to indicate explicitly the "snow grain size" is not an optical one.

P. 24, Table 3: For air temperature, cloud cover, snow depth, snow patchiness, snow temperature, soil surface temperature, and snow water content, please indicate the types of sensors (as well as manufacturer and location of the manufacturer) used to measure these properties. Regarding IceCube and Snow Fork, please indicate manufacturers and their locations. Also, please explain how the authors measured impurities in snow [%]?

––––––––––––––––––––––––––––––––––––––––––––––––––––––––––––––––––––––––

Technical corrections

P. 8, L. 207: "Analytical Spectral Devices" -> "ASD"; already defined.

P. 15, Figure 5: Consider rephrasing "Wet snow with littered surface" to "Wet snow with forest litters".

P. 16, L. 266: "Jr" -> "JR"

Sections 4 and 5 should be merged, then, please consider add some subsections.

P. 25, Equation (3): "L_{sN}" -> "L_{sN} (\lambda )"

P. 25, L. 453: "L_{s1} and L_{s2}" should be "L_{sN}"?

––––––––––––––––––––––––––––––––––––––––––––––––––––––––––––––––––––––––

References

AMAP: Snow, Water, Ice and Permafrost in the Arctic (SWIPA) 2017, Arctic Monitoring

and Assessment Programme (AMAP), Oslo, Norway. xiv + 269 pp, 2017.

Aoki, T., Aoki, T., Fukabori, M., Hachikubo, A., Tachibana, Y., and Nishio, F.: Effects of snow physical parameters on spectral albedo and bidirectional reflectance of snow surface, J. Geophys. Res., 105, 10219–10236, doi:10.1029/1999JD901122, 2000.

Carmagnola, C. M., Morin, S., Lafaysse, M., Domine, F., Lesaffre, B., Lejeune, Y., Picard, G., and Arnaud, L.: Implementation and evaluation of prognostic representations of the optical diameter of snow in the SURFEX/ISBA-Crocus detailed snowpack model, The Cryosphere, 8, 417–437, doi:10.5194/tc-8-417-2014, 2014.

Fierz, C., Armstrong, R. L., Durand, Y., Etchevers, P., Greene, E., McClung, D. M., Nishimura, K., Satyawali, P. K., and Sokratov, S. A.: The International Classification for Seasonal Snow on the Ground, IHP-VII Technical Documents in Hydrology N_83, IACS Contribution N_1, UNESCO-IHP, Paris, viii, 80 pp., 2009

Tanikawa, T., Hori, M., Aoki, T., Hachikubo, A., Kuchiki, K., Niwano, M., Matoba, S., Yamaguchi, S., and Stamnes, K., In-situ measurement of polarization properties of snow surface under the Brewster geometry in Hokkaido, Japan and northwest Greenland ice sheet, J. Geophys. Res. Atmos., 119, 13946–13964, doi:10.1002/2014JD022325, 2014.

---

## Author Comment (AC2) · 17 Dec 2019

GENERAL COMMENTS:

1) General comments from the referee

In this paper, the authors introduce and describe detailed spectral reflectance data for some types of snow, forest canopy, snow-on-canopy, snow-free patches, etc obtained from laboratory, field, mast-borne, and airborne optical measurement systems. The study areas are mainly located in the Arctic region of Finland. The main purpose of the data acquisition is to provide basic information for the development of new and

improved optical snow mapping methods for boreal forested area using satellite data. This kind of remote sensing study is very important recently, because seasonal snow physical conditions, which would affect water resources around the area for example, are rapidly changing due to the ongoing global warming. This reviewer found the data acquisition methods and procedures described in this paper are solid, and presented data are reliable in my opinion. Overall, this paper is detailed, well written, and structured; however, this reviewer suggests the following points to be considered before the publication.

Please note that page and line numbers are denoted by "P" and "L", respectively.

2) Author's response on general comments by the referee

We thank the anonymous referee #2 for the professional and constructive comments allowing us to further improve our manuscript. Please find below the point-by-point answers for each individual comment.

—————————————————

SPECIFIC COMMENTS (MAJOR)

1) Major comment from the referee

P. 16, L. 262: How about showing the standard deviations mentioned here in this paper? I think this information, which show accuracy of the data, is very important.

2) Author's response and 3) changes in manuscript

Yes, we will add the observed standard deviations to Figure 5. In case of pine/spruce spectra the standard deviation describes the measurement accuracy as the deviation is defined within the consequent measurement acquisitions averaged for one reflectance spectrum. In Fig. 5c the mean reflectance of several snow samples sampled from the same snow types and measured with same view-illumination geometry are presented. In this case the standard deviation describes the reflectance deviation between different snow samples collected and measured from the same snow type. Thus, the latter standard deviation does not describe the accuracy of the measurements but overall deviation in the snow type's reflectance. The latter was not clear in the text nor in the figure caption and we will clarify this in the revised manuscript.
* * *
1) Major comment from the referee

P. 16, L. 276: What do the authors think about the effects of the tripod on the measured reflectance data? Please discuss briefly here.

2) Author's response and 3) changes in manuscript

The effect of the tripod on the measured reflectance was very small. To avoid direct shading of the target by the tripod and measurement equipment, the tripod was placed towards the sun. The measured target area did not include the tripod legs. Nevertheless, the tripod with the extended arm obscured a part of the diffuse skylight illuminating the target. This effect has not been quantified nor corrected in our measurements and we will add this note to the revised manuscript.
* * *
1) Major comment from the referee

P. 20, L. 346 _ 347: Please explain more in detail about the resampling procedure (about the choice of a weighting function, etc).

2) Author's response and 3) changes in manuscript

In the resampling procedure the wavelengths corresponding to MODIS band 4 (545-565 nm) were chosen and weighted averages were calculated by using the relative spectral response function (RSR) provided by the data provider. We will add a more detailed description.

————————————————————————

SPECIFIC COMMENTS (MINOR)

1) Minor comment from the referee

P. 1, L. 16: For the location of Sodankylä Arctic Space Centre, please indicate altitude of the site together with the Lat Lon information.

2) Author's response and 3) changes in manuscript

We will add the altitude (179 m) in the revised manuscript.

————————————————————————-

1) Minor comment from the referee

P. 2, L. 33: Maybe, citing the latest version of the SWIPA report (AMAP, 2017) instead of the previous report (AMAP, 2011) would be better here.

2) Author's response and 3) changes in manuscript

Thank you for the remark. We will change the citation to the latest version.

————————————————————————

1) Minor comment from the referee

P. 2, L. 46: It is better to explain the definition of "spectral endmembers" here especially for non-specialist readers.

2) Author's response and 3) changes in manuscript

Often a satellite image pixel contains several surface types, e.g. both snow-covered areas and snow-free areas during spring snow melting period. The observed reflectance value of that pixel is thus a mixture of reflective properties of the surfaces present within the pixel area (or even the surfaces in the adjacent pixels). Spectral endmember is referring to 'pure' reflectance spectra of a distinct surface type such as distinct type of

snow or tree species. Assuming, that the reflectance value of a pixel is linear/nonlinear combination of the spectral signatures of the endmembers (i.e. surface types) present within the pixel, the snow cover area can be retrieved by using inverse model based or spectral unmixing methods. The success of the characterization of the spectral behavior of the endmembers affects the amount of error and uncertainty in the final snow cover retrievals. We will add a short explanation to the manuscript.
* * *
1) Minor comment from the referee

P. 2, L. 50 _ 51: Please consider citing the papers by Aoki et al. (2000), Carmagnola et al, (2014), and Tanikawa et al. (2014).

2) Author's response and 3) changes in manuscript

Thank you. We will consider each of these papers.
* * *
1) Minor comment from the referee

P. 3, L. 72: What do the authors mean by "samples"?

2) Author's response and 3) changes in manuscript

In this case "samples" is referring to the individual spectral acquisitions collected and averaged for one measurement spectrum. However, the use of "sample" has not been clear nor consistent in the manuscript and we will clarify this in the revised version.
* * *
1) Minor comment from the referee

P. 3, L. 95 _ 97: Please indicate spectral resolutions of these data here.

2) Author's response and 3) changes in manuscript

We will add the spectral resolutions.

―――――――――――――――――――――――

1) Minor comment from the referee

P. 4, L. 111: What is fjell?

2) Author's response and 3) changes in manuscript

Fjell should have actually been fell and the word is referring to high and barren arctic hills typical for Scandinavian uplands. They can reach altitudes over 500 m and are usually dome shaped with little vegetation cover. We will an explanation of "arctic hill" (or similar) to be more descriptive.

―――――――――――――――――――――――

1) Minor comment from the referee

P. 5, L. 131: What do the authors mean by "upper atmosphere's perspective"?

2) Author's response and 3) changes in manuscript

Sodankylä is located above the Arctic Circle and as stated by Kangas et al. (2016), the region can be classified as continental sub-Arctic or boreal taiga climate by Köppen classification. However, with regard to stratospheric meteorology, Sodankylä can be classified as an Arctic site, which is often located beneath the middle or the edge of the stratospheric polar vortex. However, we will remove this expression from the revised manuscript as it is not clear.

―――――――――――――――――――――――

1) Minor comment from the referee

P. 6, L. 134: For snow classification, it is better to refer the international snow classification (Fierz et al., 2009).

2) Author's response and 3) changes in manuscript

Referee #1 also pointed out the limitations of the cited snow classification systems in respect of the snow type in Sodankylä. As the international snow classification from Fierz et al. (2009) is rather concentrating on snow classification of snow physical properties than classification from the climatic point of view (which was the point in our manuscript) we will leave out the cited references and rather rely on our own expertise to describe the snow in the Sodankylä region as suggested by the referee #1.
* * *
1) Minor comment from the referee

P. 6, L. 148: Please detail more about the white reference standard used in this study (e.g., manufacturer, location of the manufacturer, and type).

2) Author's response and 3) changes in manuscript

We will add a more detailed description of the reference standard used in the study. The standard was a white Spectralon reference plate (12.7 cm, Labsphere, USA) made of packed sintered polytetrafluoroethylene (PTFE) powder.
* * *
1) Minor comment from the referee

P. 9, L. 211 _ 213: Maybe, referring to photos in Figures 4 and 6 here would be very helpful for readers.

2) Author's response and 3) changes in manuscript

We will refer to the figures as suggested.
* * *
1) Minor comment from the referee

[Figure]

P. 10, Table. 1: I think there is a higher-resolution version of ASD Field Spec Pro. Do the authors think using a standard version of Field Spec Pro is enough for the purpose of this study (the purpose of the data is summarized well in the second paragraph of the Introduction section)?

2) Author's response and 3) changes in manuscript

The standard version of ASD Field Spec Pro (also used in this study) has a spectral resolution of 3 nm @ 700 nm and 10-12 nm @ 1400/2100 nm. The spectral sampling is 1.4 nm (350-1000 nm) and 1.1 nm (1001-2500 nm). The high-resolution version of ASD Field Spec has the same spectral resolution (3 nm) in the VIS-NIR range but resolution of 6-8 nm (depending on model) at longer wavelengths. The spectral sampling interval is the same. As most of the optical satellite instruments have band widths of 10 nm at narrowest in VIS region and 20+ nm in VIS-SWIR region we consider that the resolution of the standard version of Field spec Pro is enough for the purpose of this study. However, we acknowledge the development of hyperspectral imaging (both on-board satellites and for in-situ field studies) and thus understand that while the spectral resolution of this data record remains to be sufficient for the current purpose of the study (mostly related to optical snow cover mapping) it may not be sufficient for other purposes or for the next generation satellite sensors with higher spectral resolutions.
* * *
1) Minor comment from the referee

P. 10, Table. 1: This reviewer is interested in how the authors determined the distances from the sensors to targets especially for the Lab and Portable cases. Please explain.

2) Author's response and 3) changes in manuscript

In case of snow samples measured in the laboratory a panel with known height was placed on top of the snow sample holder. The snow sample holder, in turn, was placed on an adjustable table. The table height was adjusted so that the distance between the

tip of the measurement head and the panel (+ panel height) was 25 cm. In the case of the pine and spruce twigs measured in the laboratory the distance was approximately 25 cm when measured between the uppermost limit of twigs and the measurement head. In the field, the distance between the measurement head and the target (snow surface/ground) were measured with a ruler.
* * *
1) Minor comment from the referee

P. 11, Table 2 (The following comment is related to the previous comment for "P. 3, L. 72"): If I understand the meaning of "samples" correctly, I would like to know why the numbers of samples per target in Table 2 vary from date to date. I would make the instead number the same throughout the study period when I do this kind of measurements.

2) Author's response and 3) changes in manuscript

The datasets have partially been collected within different projects/campaigns and thus the number of samples (i.e. the number of consequent spectra collected and averaged to represent one target spectrum) is not constant throughout the data record. The number of samples chosen in each case is supposed to be the best decision for those measurement conditions and for those targets. For the snow measurements in laboratory when newly precipitated snow was measured in 2015 the number of spectra per sample were reduced from previous years as this kind of snow is very sensitive to metamorphism; it was noticed that the number of consequent measurement acquisitions collected during previous experiments was not the most optimal for other types of snow and thus this number was changed.
* * *
1) Minor comment from the referee

P. 19, L. 317: Are "hours 10, 12, and 14" in local time?

2) Author's response and 3) changes in manuscript

The hours are in UTC time. We will indicate this in the revised version.

————————————————————-

1) Minor comment from the referee

P. 22, Figure 10a: Please mention where the sensor is attached in the helicopter.

2) Author's response and 3) changes in manuscript

The AISA sensor was attached in a box mounted on the bottom of the helicopter. In the bottom of the box was a hole for the sensor and the instrument foreoptics unit was set to look at nadir (0âŮę) direction. We will add this to the figure caption.

————————————————————-

1) Minor comment from the referee

P. 24, L. 419 and Table 3: Please consider adding "geometric" before "snow grain size" to indicate explicitly the "snow grain size" is not an optical one.

2) Author's response and 3) changes in manuscript

We will add "geometric" to the text and to the Table 3 to indicate that the grain size is referring to the 'traditional' snow grain size.

——————————————————————

1) Minor comment from the referee

P. 24, Table 3: For air temperature, cloud cover, snow depth, snow patchiness, snow temperature, soil surface temperature, and snow water content, please indicate the types of sensors (as well as manufacturer and location of the manufacturer) used to measure these properties. Regarding IceCube and Snow Fork, please indicate manufacturers and their locations. Also, please explain how the authors measured impurities

in snow [%]?

2) Author's response and 3) changes in manuscript

We will add the sensors and manufacturers where necessary. The snow patchiness was visually estimated by the measurer when observing the surrounding area. Also cloud cover was estimated only visually. The manufacturers and their locations for IceCube and Snow Fork will also be added. The mark for amount of impurities in snow (%) was a mistake and only whether there was forest litter visible in the snow surface have been indicated with a number. We will also correct this in the revised manuscript.
* * *
TECHNICAL CORRECTIONS

1) Technical correction from the referee

P. 8, L. 207: "Analytical Spectral Devices" -> "ASD"; already defined.

2) Author's response and 3) changes in manuscript

Thank you. We will correct this.
* * *
1) Technical correction from the referee

P. 15, Figure 5: Consider rephrasing "Wet snow with littered surface" to "Wet snow with forest litters".

2) Author's response and 3) changes in manuscript

We will rephrase the legend as suggested.
* * *
1) Technical correction from the referee

[Figure]

P. 16, L. 266: "Jr" -> "JR"

2) Author's response and 3) changes in manuscript

Will be corrected.
* * *
1) Technical correction from the referee

Sections 4 and 5 should be merged, then, please consider add some subsections.

2) Author's response and 3) changes in manuscript

Thank you. We will merge these sections and add some subsections.
* * *
1) Technical correction from the referee

P. 25, Equation (3): "L_{sN}" -> "L_{sN} (nlambda )"

2) Author's response and 3) changes in manuscript

'(lambda )' will be added.
* * *
1) Technical correction from the referee

P. 25, L. 453: "L_{s1} and L_{s2}" should be "L_{sN}"?

2) Author's response and 3) changes in manuscript

We will change this in the revised manuscript.
* * *
REFERENCES

AMAP: Snow, Water, Ice and Permafrost in the Arctic (SWIPA) 2017, Arctic Monitoring

and ssessment Programme (AMAP), Oslo, Norway. xiv + 269 pp, 2017.

Aoki, T., Aoki, T., Fukabori, M., Hachikubo, A., Tachibana, Y., and Nishio, F.: Effects of snow physical parameters on spectral albedo and bidirectional reflectance of snow surface, J. Geophys. Res., 105, 10219–10236, doi:10.1029/1999JD901122, 2000.

Carmagnola, C. M., Morin, S., Lafaysse, M., Domine, F., Lesaffre, B., Lejeune, Y., Picard, G., and Arnaud, L.: Implementation and evaluation of prognostic representations of the optical diameter of snow in the SURFEX/ISBA-Crocus detailed snowpack model, The Cryosphere, 8, 417–437, oi:10.5194/tc-8-417-2014, 2014.

Fierz, C., Armstrong, R. L., Durand, Y., Etchevers, P., Greene, E., McClung, D. M.,Nishimura, K., Satyawali, P. K., and Sokratov, S. A.: The International Classification for Seasonal Snow on the Ground, IHP-VII Technical Documents in Hydrology N_83, IACS Contribution N_1, UNESCO-IHP, Paris, viii, 80 pp., 2009

Tanikawa, T., Hori, M., Aoki, T., Hachikubo, A., Kuchiki, K., Niwano, M., Matoba, S., Yamaguchi, S., and Stamnes, K., In-situ measurement of polarization properties of snow surface under the Brewster geometry in Hokkaido, Japan and northwest Greenland ice sheet, J. Geophys. Res. Atmos., 119, 13946–13964, doi:10.1002/2014JD022325, 2014.

————————————————-

References by the authors:

Kangas, M., Rontu, L., Fortelius, C., Aurela, M., and Poikonen, A.: Geosci. Instrum. Method. Data Syst., 5, 75-84, 2016.

---

## Author Response (AR1)

Author's response to interactive comments by the anonymous Referee #1 on "Laboratory, field, mast-borne and airborne spectral reflectance measurements of boreal landscape during spring" by Henna-Reetta Hannula et al.

**GENERAL COMMENTS**

**1) General comments from the referee**

The paper describes the data collection of spectral reflectances conducted mainly in the Sodankylä region over multiple platforms. The measurements aim to be representative of different snow and weather conditions (e.g. wet or dry snow for the former, clear or covered for the latter) and were organised such that measurements from different platforms were overlapping.

The manuscript is well written and easy to read from start to finish. The laboratory measurements, field campaigns and instrumentation as well as potential sources of errors and uncertainties are thoroughly described. The multi-platform measurements seem to have been very well coordinated and organised. As a consequence I would change very little of what is currently in the manuscript and only have minor comments/clarifications.

My two main comments therefore do not concern what is in the manuscript, but what is missing from it. Firstly, one of the most interesting aspect of this study is the availability of data looking at the spectral reflectance of the same surfaces but with different instruments. The manuscript incomplete and will remain so unless a section (1) compares the reflectances obtained on different platforms on overlapping dates over the same surfaces (2) discusses the implications of the differences, bearing in mind future users (3) plots from multiple platforms showing spectral reflectances on overlapping days over the same surfaces are added.

Secondly, the data in zenodo are well organised, but their (justified) discretization into platform and scale means there is a large number of files for potential future users to wade through. It would be useful, for each platform and scale, to include representative plots of each dataset in zenodo to have a quick visualisation of the sort of data available. It is one thing to make data available, it is another to make them user-friendly and, as such, re-usable. The manuscript describes invaluable datasets that should be published and used, and I trust that adding such quick visualisation of the data through these plots will help make these datasets more user-friendly.

I trust that the manuscript will be fit to publish when the above suggestions and minor comments below are addressed.

**2) Author's response and 3) changes in manuscript**

We thank the referee #1 for a thorough reading of the manuscript and bringing out the current weaknesses. We agree that without comparison of reflectance from different platforms (from overlapping times and same surface targets) the possibly most valuable part of the data record remains underemphasized. Accordingly, we think that adding some plots of data on Zenodo would benefit the user by giving an instant idea of the dataset content.

According to the suggestions by the referee, we have added figure 13 which compares reflectance spectra of similar targets observed by different platforms. We have also added some discussion (chapter 4.2) related to implications of the different scales. Although, this discussion is not exhaustive, we believe it will raise some thoughts in the minds of the end users.

We have also added data example plots for each platform in Zenodo to give the user immediate idea of the sort of data available. Because the only way to carry this out was to add the plots as extra files, a new version from each dataset needed to be created which also resulted in a new digital identifier numbers (numbers updated in the revised manuscript).

The point-by-point answers for each of the more detailed comments can be found below. Please note that the indicated line and figure numbers are referring to the revised version (changes marked up).

**MINOR COMMENTS**

**1) Minor comment from the referee**

Minor comments: Line 134 - Sodankylä is most probably taiga snow. The Sturm snow cover classification system has been accepted as the standard in our field for a long time, but it is perhaps time we acknowledge its limitations: the European Alps are, after all, classified exclusively as maritime. While it is not the task of this manuscript, I am confident the authors are very familiar with the type of snow in Sodankylä and could therefore rely on their own expertise, rather than on a classification relying exclusively on measurements from Alaska, to describe it.

**2) Author's response and 3) changes in manuscript**

We have now relied on our own knowledge and have removed the references of snow classifications from the revised manuscript. [line 136]

**1) Minor comment from the referee**

L149 - Could there be a quick explanation of what Spectralon is?

**2) Author's response and 3) changes in manuscript**

**1) Minor comment from the referee**

L395 - Minor difference, but I think changing the start of the sentence to "As an example, Figure 11 shows reflectance values on 5 May 2011 observed over et." would make it sound less like Fig 11 is a random example not even used in the campaign. Adding the exact date to Figure 11, rather than just "May 2011", would also help clarify.

**2) Author's response and 3) changes in manuscript**

We have now reformulated the sentence as suggested. [line 413, Figure 12]

**1) Minor comment from the referee**

Table 2: This is a big table and it is easy to lose some information. If possible, a Gantt chart or something similar showing the multiple platforms and overlapping dates would make it easier to see which measurements from which platforms are overlapping.

**2) Author's response and 3) changes in manuscript**

We admit the Table 2 is large. However, we would like to keep it to have all the information from different platforms in the same place. In addition, we have compiled a Gantt chart as suggested to easily see the overlap and time ranges of the different measurements. [Figure 4]

**1) Minor comment from the referee**

Figure 9: Is the label on the y axis correct? These are not MODIS band 4 reflectance measurements, but mast-spectroradiometer measurements to match MODIS band 4. This should be clearer.

**2) Author's response and 3) changes in manuscript**

The y-axis label truly was misleading. We have now changed it to be more descriptive. [Figure 10]

**1) Minor comment from the referee**

Conclusion: Data from Sodankylä are also being used for driving and evaluating snow models (Essery et al., 2016, gi-5-219-2016) and Earth System Models, notably as part of ESM-SnowMIP (Menard et al., 2019, essd-11-865-2019). It may be worth mentioning that adding albedo measurements to these datasets would be invaluable to the snow modelling community.

**2) Author's response and 3) changes in manuscript**

This is a good point and we have now added this in the discussion section. [lines 606-609]

Author's response to interactive comments by the anonymous Referee #2 on "Laboratory, field, mast-borne and airborne spectral reflectance measurements of boreal landscape during spring" by Henna-Reetta Hannula et al.

**GENERAL COMMENTS:**

**1) General comments from the referee**

In this paper, the authors introduce and describe detailed spectral reflectance data for some types of snow, forest canopy, snowon-canopy, snow-free patches, etc obtained from laboratory, field, mast-borne, and airborne optical measurement systems. The study areas are mainly located in the Arctic region of Finland. The main purpose of the data acquisition is to provide basic information for the development of new and improved optical snow mapping methods for boreal forested area using satellite data. This kind of remote sensing study is very important recently, because seasonal snow physical conditions, which would affect water resources around the area for example, are rapidly changing due to the ongoing global warming. This reviewer found the data acquisition methods and procedures described in this paper are solid, and presented data are reliable in my opinion. Overall, this paper is detailed, well written, and structured; however, this reviewer suggests the following points to be considered before the publication.

Please note that page and line numbers are denoted by "P" and "L", respectively.

**2) Author's response on general comments by the referee**

We thank the anonymous referee #2 for the professional and constructive comments allowing us to further improve our manuscript. Please find below the point-by-point answers for each individual comment. Note that the indicated line and figure numbers are referring to the revised version (changes marked up).

**SPECIFIC COMMENTS (MAJOR)**

**1) Major comment from the referee**

P. 16, L. 262: How about showing the standard deviations mentioned here in this paper? I think this information, which show accuracy of the data, is very important.

**2) Author's response and 3) changes in manuscript**

We have now added the standard deviations on Figure 6. In case of pine/spruce spectra the standard deviation describes the measurement accuracy as the deviation is defined within the consequent measurement acquisitions averaged for one reflectance spectrum. These standard deviations are so small that they are practically invisible in the figure 6a and 6b. In Fig. 6c the mean reflectance of several snow samples sampled from the same snow types and measured with same view-illumination geometry are presented and the added standard deviation describes the deviation between different snow samples collected and measured from the same snow type. Thus, this standard deviation does not describe the accuracy of the measurements. The latter was not clear in the text nor in the figure caption and we have now clarified this in both. [Figure 6, lines 271-275]

**1) Major comment from the referee**

P. 16, L. 276: What do the authors think about the effects of the tripod on the measured reflectance data? Please discuss briefly here.

**2) Author's response and 3) changes in manuscript**

The effect of the tripod on the measured reflectance was very small. To avoid direct shading of the target by the tripod and measurement equipment, the tripod was placed towards the sun. The measured target area did not include the tripod legs. Nevertheless, the tripod with the extended arm obscured a part of the diffuse skylight illuminating the target. This effect has not been quantified nor corrected in our measurements. This comment has been added into chapter 4.1 discussing measurement error and uncertainties. [lines 512-513]

**1) Major comment from the referee**

P. 20, L. 346 \_ 347: Please explain more in detail about the resampling procedure (about the choice of a weighting function, etc).

**2) Author's response and 3) changes in manuscript**

In the resampling procedure the wavelengths corresponding to MODIS band 4 (545-565 nm) were chosen and weighted averages were calculated by using the relative spectral response function (RSR) provided by the data provider. We have now added a more detailed description. [lines 369-371]

**SPECIFIC COMMENTS (MINOR)**

**1) Minor comment from the referee**

P. 1, L. 16: For the location of Sodankylä Arctic Space Centre, please indicate altitude of the site together with the Lat Lon information.

**2) Author's response and 3) changes in manuscript**

We have now added the altitude (179 m) in the revised manuscript. [lines 16 and 124]

**1) Minor comment from the referee**

P. 2, L. 33: Maybe, citing the latest version of the SWIPA report (AMAP, 2017) instead of the previous report (AMAP, 2011) would be better here.

**2) Author's response and 3) changes in manuscript**

Thank you for the remark. We have now cited the latest version. [line 34]

**1) Minor comment from the referee**

P. 2, L. 46: It is better to explain the definition of "spectral endmembers" here especially for non-specialist readers.

**2) Author's response and 3) changes in manuscript**

We have now added a short explanation. Often a satellite image pixel contains several surface types, e.g. both snow-covered areas and snow-free areas during spring snow melting period. The observed reflectance value of that pixel is thus a mixture of reflective properties of the surfaces present within the pixel area (or even the surfaces in the adjacent pixels). Spectral endmember is referring to 'pure' reflectance spectra of a distinct surface type such as distinct type of snow or tree species. Assuming, that the reflectance value of a pixel is linear/nonlinear combination of the spectral signatures of the endmembers (i.e. surface types) present within the pixel, the snow cover area can be retrieved by using inverse model based or spectral unmixing methods. The success of the characterization of the spectral behavior of the endmembers affects the amount of error and uncertainty in the final snow cover retrievals. [lines 47-48]

**1) Minor comment from the referee**

P. 2, L. 50 \_ 51: Please consider citing the papers by Aoki et al. (2000), Carmagnola et al, (2014), and Tanikawa et al. (2014).

**2) Author's response and 3) changes in manuscript**

Thank you. We added references for Aoki et al. (2000) and Tanikawa et al. (2014) but thought the focus of paper Carmagnola et al. (2014) (handling snowpack modeling) was not relevant within the scope of this manuscript. [lines 52-53]

**1) Minor comment from the referee**

P. 3, L. 72: What do the authors mean by "samples"?

**2) Author's response and 3) changes in manuscript**

In this case "samples" was referring to the individual spectral acquisitions collected and averaged for one measurement spectrum. We have now unified and clarified the use of "sample" in the revised manuscript. We now use "sample" in the case of snow and pine/spruce measurements executed in the laboratory as in these cases the branches and snow had been physically separated/sampled for the measurements. In other context we now use target (referring to measurement target) and simply measurement or consequent spectral acquisitions instead of "sample" to be clearer. [line 74 and throughout the manuscript]

**1) Minor comment from the referee**

P. 3, L. 95 \_ 97: Please indicate spectral resolutions of these data here.

**2) Author's response and 3) changes in manuscript**

We have now added the spectral resolutions for both instruments in the revised manuscript. [lines 98-100]

**1) Minor comment from the referee**

P. 4, L. 111: What is fjell?

**2) Author's response and 3) changes in manuscript**

Fjell should have actually been fell and the word is referring to high and barren arctic hills typical for Scandinavian uplands. They can reach altitudes over 500 m and are usually dome shaped with little vegetation cover. We have now added an explanation of "arctic hill" in brackets to be more descriptive. [line 114]

**1) Minor comment from the referee**

P. 5, L. 131: What do the authors mean by "upper atmosphere's perspective"?

**2) Author's response and 3) changes in manuscript**

Sodankylä is located above the Arctic Circle and as stated by Kangas et al. (2016), the region can be classified as continental sub-Arctic or boreal taiga climate by Köppen classification. However, with regard to stratospheric meteorology, Sodankylä can be classified as an Arctic site, which is often located beneath the middle or the edge of the stratospheric polar vortex. However, we removed this expression from the revised manuscript and simplified the sentence. [lines 135-136]

**1) Minor comment from the referee**

P. 6, L. 134: For snow classification, it is better to refer the international snow classification (Fierz et al., 2009).

**2) Author's response and 3) changes in manuscript**

Referee #1 also pointed out the limitations of the cited snow classification systems in respect of the snow type in Sodankylä. As the international snow classification from Fierz et al. (2009) is rather concentrating on snow classification of snow physical properties than classification from the climatic point of view (which was the point in our manuscript) we will leave out the cited references and rather rely on our own expertise to describe the snow in the Sodankylä region as suggested by the referee #1. [line 136]

**1) Minor comment from the referee**

P. 6, L. 148: Please detail more about the white reference standard used in this study (e.g., manufacturer, location of the manufacturer, and type).

**2) Author's response and 3) changes in manuscript**

We have now added a more detailed description of the reference standard used in the study. The standard was a white Spectralon reference plate (12.7 cm, Labsphere, USA) made of packed sintered polytetrafluoroethylene (PTFE) powder. [lines 151-156]

**1) Minor comment from the referee**

P. 9, L. 211 \_ 213: Maybe, referring to photos in Figures 4 and 6 here would be very helpful for readers.

**2) Author's response and 3) changes in manuscript**

We have added the references in the revised manuscript for both figures as suggested. [lines 219-220, Figures 5 and 7]

**1) Minor comment from the referee**

P. 10, Table. 1: I think there is a higher-resolution version of ASD Field Spec Pro. Do the authors think using a standard version of Field Spec Pro is enough for the purpose of this study (the purpose of the data is summarized well in the second paragraph of the Introduction section)?

**2) Author's response and 3) changes in manuscript**

The standard version of ASD Field Spec Pro (also used in this study) has a spectral resolution of 3 nm @ 700 nm and 10-12 nm @ 1400/2100 nm. The spectral sampling is 1.4 nm (350-1000 nm) and 1.1 nm (1001-2500 nm). The high-resolution version of ASD Field Spec has the same spectral resolution (3 nm) in the VIS-NIR range but resolution of 6-8 nm (depending on model) at longer wavelengths. The spectral sampling interval is the same. As most of the optical satellite instruments have band widths of 10 nm at narrowest in VIS region and 20+ nm in VIS-SWIR region we consider that the resolution of the standard version of Field spec Pro is enough for the purpose of this study. However, we acknowledge the development of hyperspectral imaging (both on-board satellites and for in-situ field studies) and thus understand that while the spectral resolution of this data record remains to be sufficient for the current purpose of the study (mostly related to optical snow cover mapping) it may not be sufficient for other purposes or for the next generation satellite sensors with higher spectral resolutions.

**1) Minor comment from the referee**

P. 10, Table. 1: This reviewer is interested in how the authors determined the distances from the sensors to targets especially for the Lab and Portable cases. Please explain.

**2) Author's response and 3) changes in manuscript**

In case of snow samples measured in the laboratory a panel with known height was placed on top of the snow sample holder. The snow sample holder, in turn, was placed on an adjustable table. The table height was adjusted so that the distance between the tip of the measurement head and the panel (+ panel height) was 25 cm. In the case of the pine and spruce twigs measured in the laboratory the distance was approximately 25 cm when measured between the uppermost limit of twigs and the measurement head. In the field, the distance between the measurement head and the target (snow surface/ground) were measured with a ruler. [Table 1 and lines 257-258 and 297-298]

**1) Minor comment from the referee**

P. 11, Table 2 (The following comment is related to the previous comment for "P. 3, L. 72"): If I understand the meaning of "samples" correctly, I would like to know why the numbers of samples per target in Table 2 vary from date to date. I would make the instead number the same throughout the study period when I do this kind of measurements.

**2) Author's response and 3) changes in manuscript**

The datasets have partially been collected within different projects/campaigns and thus the number of samples (i.e. the number of consequent spectra collected and averaged to represent one target spectrum) is not constant throughout the data record.

The number of samples chosen in each case is supposed to be the best decision for those measurement conditions and for those targets. For the snow measurements in laboratory when newly precipitated snow was measured in 2015 the number of spectra per sample were reduced from previous years as this kind of snow is very sensitive to metamorphism; it was noticed that the number of consequent measurement acquisitions collected during previous experiments was not the most optimal for other types of snow and thus this number was changed. We have slightly changed the headers in Table 2 and hope they are now clearer for the reader. [Table 2, lines 256 and 262-263]

**1) Minor comment from the referee**

P. 19, L. 317: Are "hours 10, 12, and 14" in local time?

**2) Author's response and 3) changes in manuscript**

The hours are in UTC time. We have now indicated this in the revised manuscript. [line 332]

**1) Minor comment from the referee**

P. 22, Figure 10a: Please mention where the sensor is attached in the helicopter.

**2) Author's response and 3) changes in manuscript**

The AISA sensor was attached in a box mounted on the bottom of the helicopter. In the bottom of the box was a hole for the sensor and the instrument foreoptics unit was set to look at nadir  $(0^\circ)$  direction. [Figure 11]

**1) Minor comment from the referee**

P. 24, L. 419 and Table 3: Please consider adding "geometric" before "snow grain size" to indicate explicitly the "snow grain size" is not an optical one.

**2) Author's response and 3) changes in manuscript**

We have added "geometric" to the text and to the Table 3 to indicate that the grain size is referring to the 'traditional' snow grain size. [Table 3 and line 437]

**1) Minor comment from the referee**

P. 24, Table 3: For air temperature, cloud cover, snow depth, snow patchiness, snow temperature, soil surface temperature, and snow water content, please indicate the types of sensors (as well as manufacturer and location of the manufacturer) used to measure these properties. Regarding IceCube and Snow Fork, please indicate manufacturers and their locations. Also, please explain how the authors measured impurities in snow [%]?

**2) Author's response and 3) changes in manuscript**

We have added the sensors and manufacturer where necessary. The manufacturer for the thermometer used in the temperature measurements in the portable field observations is not known and could not be recovered from the measurement documents but it is a corresponding digital thermometer to the TH310 utilized in the snow laboratory experiments. The snow patchiness was visually estimated by the measurer when observing the surrounding area. Also cloud cover was estimated only visually. The manufacturers and their locations for IceCube and Snow Fork have also been added. The mark for amount of impurities in snow (%) was a mistake and only whether there was forest litter visible in the snow surface have been indicated with a number. We have now clarified this in the revised manuscript. [Table 3, line 438]

**TECHNICAL CORRECTIONS**

**1) Technical correction from the referee**

P. 8, L. 207: "Analytical Spectral Devices" -> "ASD"; already defined.

**2) Author's response and 3) changes in manuscript**

Thank you. We have corrected this. [line 213]

**1) Technical correction from the referee**

P. 15, Figure 5: Consider rephrasing "Wet snow with littered surface" to "Wet snow with forest litters".

**2) Author's response and 3) changes in manuscript**

We have rephrased the legend entry in the revised manuscript. [Figure 6]

**1) Technical correction from the referee**

P. 16, L. 266: "Jr" -> "JR"

**2) Author's response and 3) changes in manuscript**

Have been corrected. [line 282]

**1) Technical correction from the referee**

Sections 4 and 5 should be merged, then, please consider add some subsections.

**2) Author's response and 3) changes in manuscript**

Thank you. We have now merged sections 4 and 5 in the revised manuscript and have also added a subsection shortly discussing the implications of scale when reflectance values of the same targets but measured with different platforms are compared (comment from referee #1). [Section 4]

**1) Technical correction from the referee**

P. 25, Equation (3): "L\_{sN}" -> "L\_{sN} (nlambda )"

**2) Author's response and 3) changes in manuscript**

'(lambda )' have been added. [Equation 3]

**1) Technical correction from the referee**

P. 25, L. 453: "L\_{s1} and L\_{s2}" should be "L\_{sN}"?

**2) Author's response and 3) changes in manuscript**

We have now changed this in the revised manuscript. [line 471]

1Space and Earth Observation Centre, Finnish Meteorological Institute, Helsinki, FI-00560, Finland 2Geoinformatics Research, Finnish Environment Institute, Helsinki, FI-00790, Finland 3Program for Environmental Information, Finnish Environment Institute, Helsinki, FI-00790, Finland

Correspondence to: Henna-Reetta Hannula (Henna-Reetta.Hannula@fmi.fi)

5

- 10 Abstract. We publish and describe a surface spectral reflectance data record of seasonal snow (dry, wet, shadowed), forest ground (lichen, moss) and forest canopy (spruce and pine, branches) constituting the main elements of the boreal landscape. The reflectances are measured with spectro(radio)meters covering the wavelengths from visible (VIS) to short-wave infrared (SWIR) (350 to 2500 nm). In this paper, we describe the instruments used and how the spectral observations at different scales along with the concurrent in situ reference data have been collected, processed and archived. Information on the quality of the
- 15 data and factors causing uncertainty are discussed. The main experimental site is located in Sodankylä Arctic Space Centre in northern Finland (67.37° N, 26.63° E, 179 m.a.s.l) and the surrounding region. The collection includes highly controlled snow and conifer branch laboratory spectral measurements, portable field spectroradiometer observations of snow and snow-free ground at different locations and continuous mast-borne reflectance time series data of a pine forest and forest opening. In addition to the surface level spectral reflectance, data from airborne imaging spectrometer campaigns over Sodankylä boreal
- 20 forest and Saariselkä fell region at selected spectral bands are included in the collection. All measurements of the data record correspond to a typical polar orbiting satellite observation event in high latitude spring season regarding their sun or illumination source (calibrated lamp) zenith angle and close to nadir instrument viewing angle. For all measurement geometries, observations are given in surface reflectance quantity corresponding to the typical representation of a satellite observation quantity to facilitate their comparison with other data sources. The openly accessible spectral reflectance data at
- 25 multiple scales are suitable e.g. to climate and hydrological research and remote sensing model validation and development. To facilitate easy access to the data record the four datasets described here are deposited in a permanent data repository (http://www.zenodo.org/communities/boreal\_reflectances/) (Hannula et al., 2019). Each dataset of a distinct scale has its own unique DOI (laboratory: 10.5281/zenodo.26774773580078) (Hannula and Heinilä, 2018a), field: 10.5281/zenodo.26536293580825 (Heinilä et al., 2019a), mast-borne: 10.5281/zenodo.33497473580096 (Hannula and

Heinilä, 2018b), airborne: 10.5281/zenodo.30484203580451 (Heinilä, 2019a), and 10.5281/zenodo.3580481902 (
[revised manuscript text omitted]